# Goal-directed and flexible modulation of syllable sequence within birdsong

Takuto Kawaji [1], Mizuki Fujibayashi [1] & Kentaro Abe [1,2] ✉

Songs constitute a complex system of vocal signals for inter-individual communication in songbirds. Here, we elucidate the flexibility which songbirds exhibit in the organizing and sequencing of syllables within their songs. Utilizing a newly devised song decoder for quasi-real-time annotation, we execute an operant conditioning paradigm, with rewards contingent upon specific syllable syntax. Our analysis reveals that birds possess the capacity to modify the contents of their songs, adjust the repetition length of particular syllables and employing specific motifs. Notably, birds altered their syllable sequence in a goal-directed manner to obtain rewards. We demonstrate that such modulation occurs within a distinct song segment, with adjustments made within 10 minutes after cue presentation. Additionally, we identify the involvement of the parietal-basal ganglia pathway in orchestrating these flexible modulations of syllable sequences. Our findings unveil an unappreciated aspect of songbird communication, drawing parallels with human speech.

Songs of songbirds comprise vocal elements, referred to as syllables, which are strung together in sequences. Despite the primary role of these songs as a mode of communication, a predominant hypothesis suggests that the arrangement of syllables within songs lacks substantial content, including semantic information[1,2]. This feature has been debated as a crucial difference between human speech and the songs of birds[3]. Nonetheless, certain bird species employ different songs comprising a different sequence of syllables in different circumstances[4–7]. Previous research has demonstrated that altering the order of syllables in songs or calls can elicit a different behavioral response in listeners[8–10]. These findings suggest that the order of syllables carries specific information to which the listeners respond by modifying their behaviors. However, we lack comprehensive information on whether birds can adaptively modulate the sequential order of syllables within their song to communicate with others.

Bengalese finches (*Lonchura striata* var domestica) use songs that are acquired by learning through social experience[11]. They utter variable sequences of songs, but the songs as a whole are stereotyped and stable[12]. Whether and how they can modulate the variability of syllable sequence remains to be revealed. Through operant conditioning utilizing online song disruption by white-noise feedback, Bengalese finches can be trained to change the phonological aspect of the target

syllable or avoid using some songs consisting of specific syllables or motifs[13–15]. However, it remains unclear whether birds can intentionally modulate the particular sequence of their songs according to internal and external conditions[16]. Specifically, whether they can modulate the specific sequence in their songs to obtain social feedback is unknown. In this work, we constructed a quasi-real-time song decoder that translated the syllable sequence of a song-bout within the latency of natural communication among birds and utilized it for operant conditioning of their songs.

## Results

### Rapid, online decoding of syllable sequences in birdsongs

First, we set up an experimental system to provide feedback signals contingently according to the contents of their songs to analyze the communicative value of information regarding syllable sequence in birdsongs. This contingent feedback, involving a closed-loop experimental operation, crucially requires rapid decoding of syllable sequences in songs. For this purpose, we developed a song decoder called simultaneous automatic interpreter for birdsongs (SAIBS), which can annotate the syllable orders within songs in a quasi-real-time manner. SAIBS performs the auto-segmentation and auto-labeling of audio files and executes fully automated syllable annotation in

[1]Lab of Brain Development, Graduate School of Life Sciences, Tohoku University, Katahira 2-1-1, Sendai, Miyagi 980-8577, Japan. [2]Division for the Establishment of Frontier Sciences of the Organization for Advanced Studies, Tohoku University, Sendai, Miyagi 980-8577, Japan. ✉e-mail: k.abe@tohoku.ac.jp

birdsongs. The automatic segmentation and clustering of syllables in songs facilitate reliable and objective annotation compared with that of manual or semi-automated annotations relying on manually segmented training data. Translating songs using SAIBS consists of an offline training phase and an online decoding phase (Fig. 1a). In the offline training phase; we set up the parameters of the decoder for each bird using the recorded audio data of the subject without manually annotating the syllables. Specifically, to identify syllables, the spectral information of each sound element in the recorded songs was compressed by the t-distributed stochastic neighbor embedding, t-SNE algorithm[17], followed by density-based spatial clustering of applications with noise (DBSCAN)[18]. Subsequently, a convolutional neural network (CNN) was used to generate a song decoder that classified each syllable and annotated them into letters (Fig. 1b). In the online decoding phase, the trained decoder detects syllables in incoming audio data and outputs each syllable annotation (Fig. 1c).

Syllable annotation using SAIBS was shown to be accurate and reproducible. Regarding the accuracy of annotation, SAIBS was significantly more accurate than manual annotation and comparable to cutting-edge song decoders such as TweetyNet[19]. We measured the accuracy of the SAIBS online decoder by comparing it with the annotation created by TweetyNet, which was trained using the annotation created by SAIBS (see Supplementary Note). The sequence identity rate of syllable annotations between TweetyNet and SAIBS was $98.43 \pm 0.28\%$ and $96.91 \pm 0.44\%$, excluding and including the introductory note, respectively (Fig. 1d). The syllable labeling match rate depended on the syllable type, ranging from 86.3–99.9% (Fig. 1e), and the rate was lowest for the syllables with rare occurrences in the sample data (syllable "n"). The SAIBS decoder was significantly more accurate than the manual annotation (four skilled researchers; $72.27 \pm 5.33\%$, $P = 4.84 \times 10^{-3}$, Welch's t-test, $t(2) = 7.45$; Supplementary Fig. 1), as manual annotation tends to be affected by individual's difference of syllable labeling. Regarding reproducibility, the annotation yield was always the same for the offline decoding of pre-recorded songs and $98.26 \pm 0.19\%$ for online analysis of ongoing recordings.

Independent training of SAIBS resulted in a sequence identity rate with a coefficient variance of 0.317%, which was significantly less than that of manual annotation, 7.38% ($P = 7.31 \times 10^{-3}$, test of difference between correlations). The score of SAIBS on other Bengalese finch gave a similar value of syllable match rate ($98.47 \pm 0.13\%$; Supplementary Fig. 2a). *SAIBS* can also be utilized for annotating songs of zebra finches, which were composed with fewer syllables with an accuracy of $98.64 \pm 0.31\%$ (Supplementary Fig. 2b, c).

The annotation using SAIBS was rapid. The time required for computing annotation was 6 ms after the end of the syllable detection, which was comparable to the rapid song decoder[20]. In our experimental setup, the actual time lag of presenting feedback using SAIBS online decoding was measured at $109 \pm 19$ ms from the end of the songs. Thus, SAIBS enables quasi-real-time feedback according to the uttered song contents within a temporal delay of the natural vocal communication of birds (feedback call responses, 200–300 ms in zebra finches[21]; learned behavioral reaction, 1000–1200 ms in Bengalese finches[22]).

## Operant conditioning of syllable repetition in songs

Vocal signals are used for communication; therefore, investigating the extent to which songs are acquired and modulated based on social cues is essential. We employed SAIBS to perform operant conditioning of Bengalese finches, enabling them to modulate the contents of their songs. Previous research has implemented online irruption of songs by delivering rapid feedback, such as overlaying white noise onto the vocalization, upon the detection of the target syllable within a song[15]. This syllable-targeted reinforcement demonstrated that birds could be conditioned to refrain from uttering a particular sequence of syllables[13,14]. In our study, we utilized SAIBS to deliver physiological signals, which are inherent in communication, contingent upon the contents of the uttered songs. In our pilot study, we found that presenting short movies of conspecific birds (Supplementary Movie 1), a virtual social signal, effectively elicited social behavior in the Bengalese finches, such as approaching the monitor, singing, and erecting the

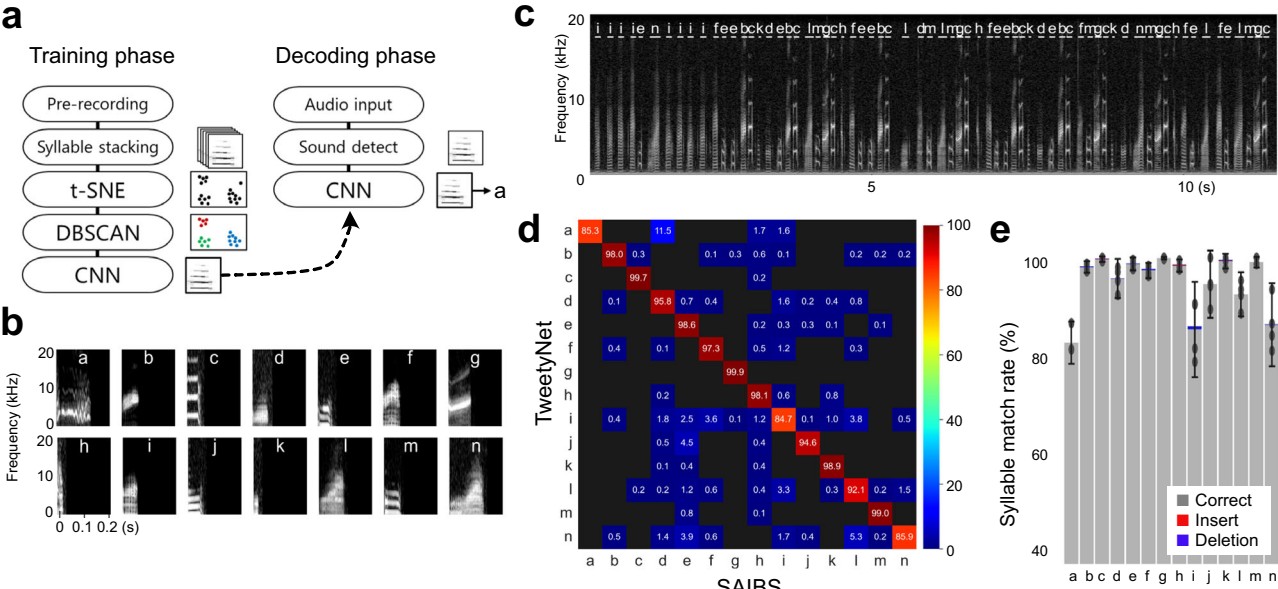

**Fig. 1 | Development and evaluation of a quasi-real-time song decoder, SAIBS. a** The operational architecture. In the training phase, syllables were clustered and used for training a convolutional neural network (CNN). In the decoding phase, trained CNN was used to decode the coming audio. t-SNE t-distributed stochastic neighbor embedding, DBSCAN density-based spatial clustering of application with noise. **b**, **c** Example of syllables automatically clustered by SAIBS (**b**) and their detection in a song (**c**). **d** Annotation comparison against TweetyNet. Songs from one bird were annotated by SAIBS, and the results were compared with those by TweetyNet. The matrix shows the mean match rate between each decoder for each syllable. **e** The concordance rate for each syllable is shown with rates of insert-type and deletion-type errors. Mean ± s.e.m. from $n = 4$ trial.

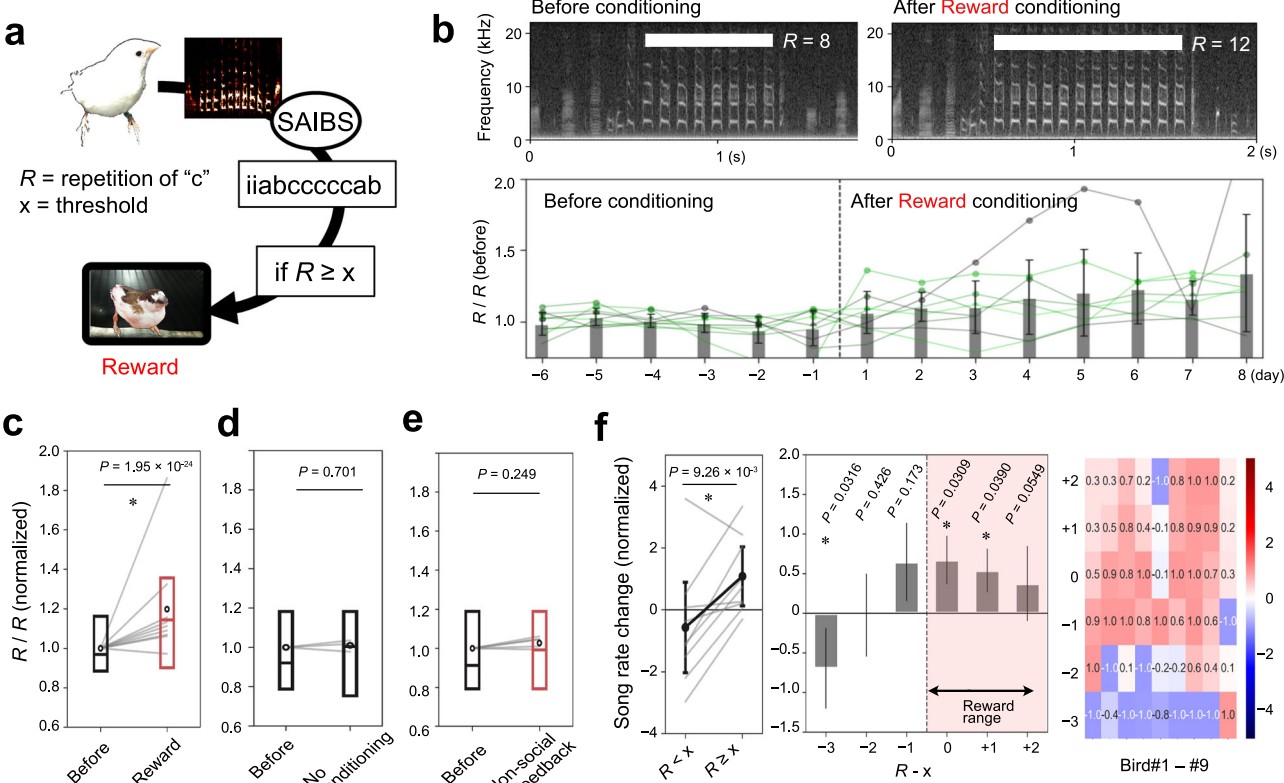

**Fig. 2 | Operant conditioning of syllable repetition. a** The repetition of the target syllables in songs was analyzed online by SAIBS. If the repetition number of the target syllable "c" exceeds the predetermined threshold "x", the reward movie was presented from a monitor. **b** Example of song spectrograms before and after the conditioning (Top). The daily shift of max repetition counts in nine birds (Below). Mean ± s.e.m. **c**–**e** Change of syllable repetition with conditioning (**c**) and without conditioning (**d**), and with conditioning with non-social feedback movie (**e**). The box plot shows median and first and third quantiles with mean shown as circles; $P$ values, two-sided paired $t$-test, $t(8) = -10.5$, $n = 9$ (**c**), $t(4) = -0.384$, $n = 5$ (**d**), and $t(4) = -1.15$, $n = 5$ (**e**). **f** Shift of song rate of songs of each repetition before and after conditioning. Pink shadings, the rewarded range. Mean ± s.e.m.; $P$ values, two-sided paired $t$-test (left, middle), $t(5) = 3.41$, $n = 6$. The summary (left, middle) and results from each bird (right) are shown. The reward range is highlighted in pink. The heatmap shows the song rate change (%) for each bird.

feathers[23]. The contingent presentation of conspecific movie have been shown to stimulate song learning in juvenile zebra finch[23]. Thus, we employed conspecific movies as reinforcers within this conditioning paradigm. We hypothesized that providing such social signals contingently upon the utterance of specific contents within the song would result in adaptive changes to the corresponding locus in their songs.

The songs of Bengalese finches differ among individuals, with many individuals containing a repetition of particular syllables in their songs. The number of repetitions in a song-bout cannot be modeled using a first-order Markov process[24,25]. First, we targeted the repetitive syllable for conditioning in order to ask whether birds can learn to modulate syllable sequence in their songs. Using SAIBS, we counted repetitive syllables, the number of which is represented as "$R$" in each uttered song, and accordingly presented the reward to the bird if the "$R$" surpassed a threshold, represented as "$x$," which was determined for each bird (Fig. 2a). After conditioning, we observed an increase in the number of syllable repetitions (Fig. 2b, c), which did not occur without conditioning (Fig. 2d). There was a significant difference in the effect between conditioning and no-conditioning ($P = 0.00497$, linear mixing effect (LME) model, $F(1799, 999) = 7.90$). We did not observe an increase in repetition when a movie with non-animate signals was used for feedback (Fig. 2e and Supplementary Movie 2). A significant difference was observed between the effects of social reward and non-social feedback ($P = 0.025$, LME model, $F(1799, 769) = 5.03$), indicating that it was the social reward, and not the contingent visual signals, that stimulated the change in song structure. Analysis of the change in the number of repetitive syllables for each bird revealed that birds

increased the number of repetitions that surpassed the threshold ($R \geq x$; Fig. 2f). Most birds increased the song bout with the repetitive syllable −1 to +2 to the threshold ($-1 \leq R - x \leq +2$), whereas they reduced songs of less than −2 repetition to the threshold ($R - x \leq -2$; Fig. 2f).

Furthermore, we assessed whether birds could modulate their behavior to change the repetition to a particular range. For some birds that extended the repetition to more than two syllables beyond the threshold, we applied a new rule, the "Not-too-much" rule, during which the reward was not provided if the birds sang a too-long repetition in their songs (Supplementary Fig. 3). This procedure corresponds to shaping in classical conditioning[26]. During the "Not-too-much" rule condition, birds were still rewarded if the repetition in a song-bout exceeded the reward threshold but were omitted if the repetition was over two repetitions above the threshold ($R > x + 1$; Supplementary Fig. 3a). After applying the "Not-too-much" rule, we observed a significant increase in repetitions within the reward range of "Not-too-much" ($x \leq R \leq x + 1$), whereas songs below the threshold ($R < x$) were reduced, and no significant difference was observed in the rate of repetition surpassing the omission threshold ($R > x + 1$; Supplementary Fig. 3c). These results demonstrate that birds could flexibly modulate the range of syllable repetition based on the reward provided.

## Song change occurs at a defined locus within the song

To further reveal the mechanism by which conditioning-dependent change of songs occurs, we analyzed the change in their syntactical structure after the conditioning. Markov transition probability analysis of syllables revealed that the overall structures of songs were

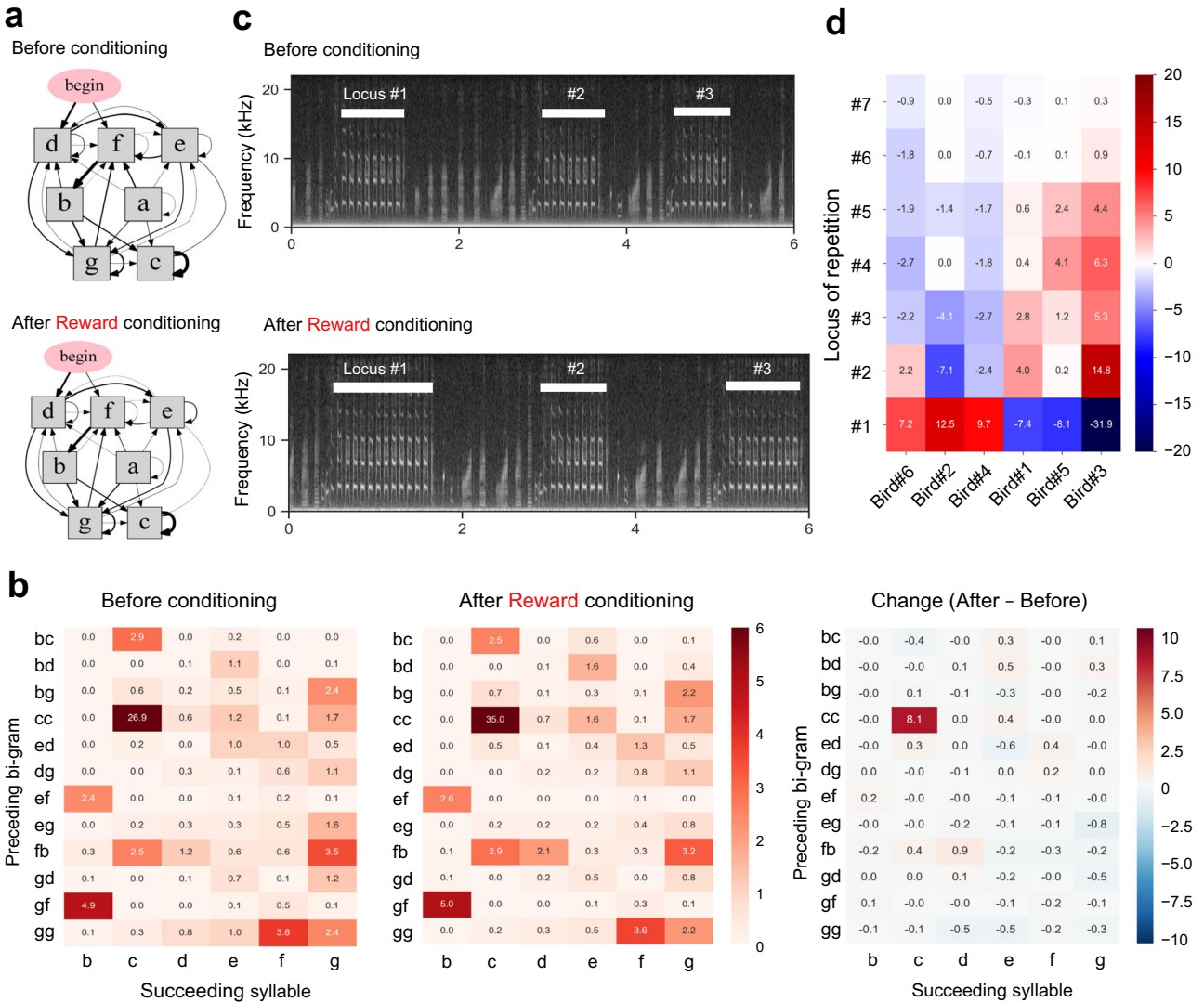

**Fig. 3 | Song change occurs at a defined locus within the song. a** Markov diagrams of syllable transition (bi-gram) before and after the conditioning of syllable repetition. **b** Tri-gram syllable transition matrix before (left) and after (middle) the conditioning, and the difference of before and after (right). This subject was conditioned to increase syllable repetition "c". **c** Spectrograms of songs before and after conditioning. The repetition occurred at three loci of single song bout in this example. **d** Change of repetition locus where the maximum repetition occurs in a song bout. The heatmap shows the song rate change (%) before and after conditioning for each bird.

preserved (Fig. 3a). This observation was further confirmed by a tri-gram analysis of the syllable transition probability (Fig. 3b), indicating a high correlation before and after conditioning (0.792 ± 0.113, Pearson's rank correlation coefficient of top 30 tri-gram, mean ± s.d., $n = 3$ birds), which is not different from the correlation of birds without conditioning (0.719 ± 0.037; $P = 0.366$, test of the difference between two correlation coefficients). For example, in bird #1, the probability of target syllable "c" after "c-c" was increased by 8.09% (1.3-fold change), but the mean change of non-target syllable transition was around 0.33% (average of top 2–30 transition, 0.90-fold change; Fig. 3b).

A song bout of Bengalese finch often contains multiple loci of repetition (Fig. 3c). In our conditioning paradigm, rewards were administered whenever the repetition at any locus surpassed the designated threshold. Thus, the specific loci of modulation was not the direct factor determining the reward outcome. Nevertheless, we observed that each bird tended to modulate repetition at a particular locus. For example, in the case of birds #2, #4, and #6, the conditioning-dependent increase in repetition occurred at the first repetitive locus. In contrast, the modulation occurred in the latter loci in birds #1, #3, and #5 (Fig. 3d). The loci of modulation varied among

the individuals, suggesting that such locus of modulation depended on the strategies of each bird. These results suggest that learning-dependent song modulation occurs within a specific location within songs rather than through the use of different songs in their repertoire.

### Context-dependent and goal-directed modulation of song contents

Next, we analyzed whether the conditioning of syllable sequence was affected by the subjective value of the reward. To devaluate the subjective value of the reward, we applied an "Excess-reward" rule during which the birds were given excessive rewards unconditionally, irrespective of their song content (Fig. 4a). After three consecutive days of "Excess-reward" conditioning, we observed a significant decrease in repetition number (Fig. 4b). This result indicates that the behavioral outcome was influenced by the subjective value of rewards and also occludes the possibility that the modulation of songs observed depended on affective responses to the reward.

The modulation of song contents was context-dependent. After the conditioning, we introduced days with a "No-reward" rule during which the birds were not rewarded, even if their repetition was long

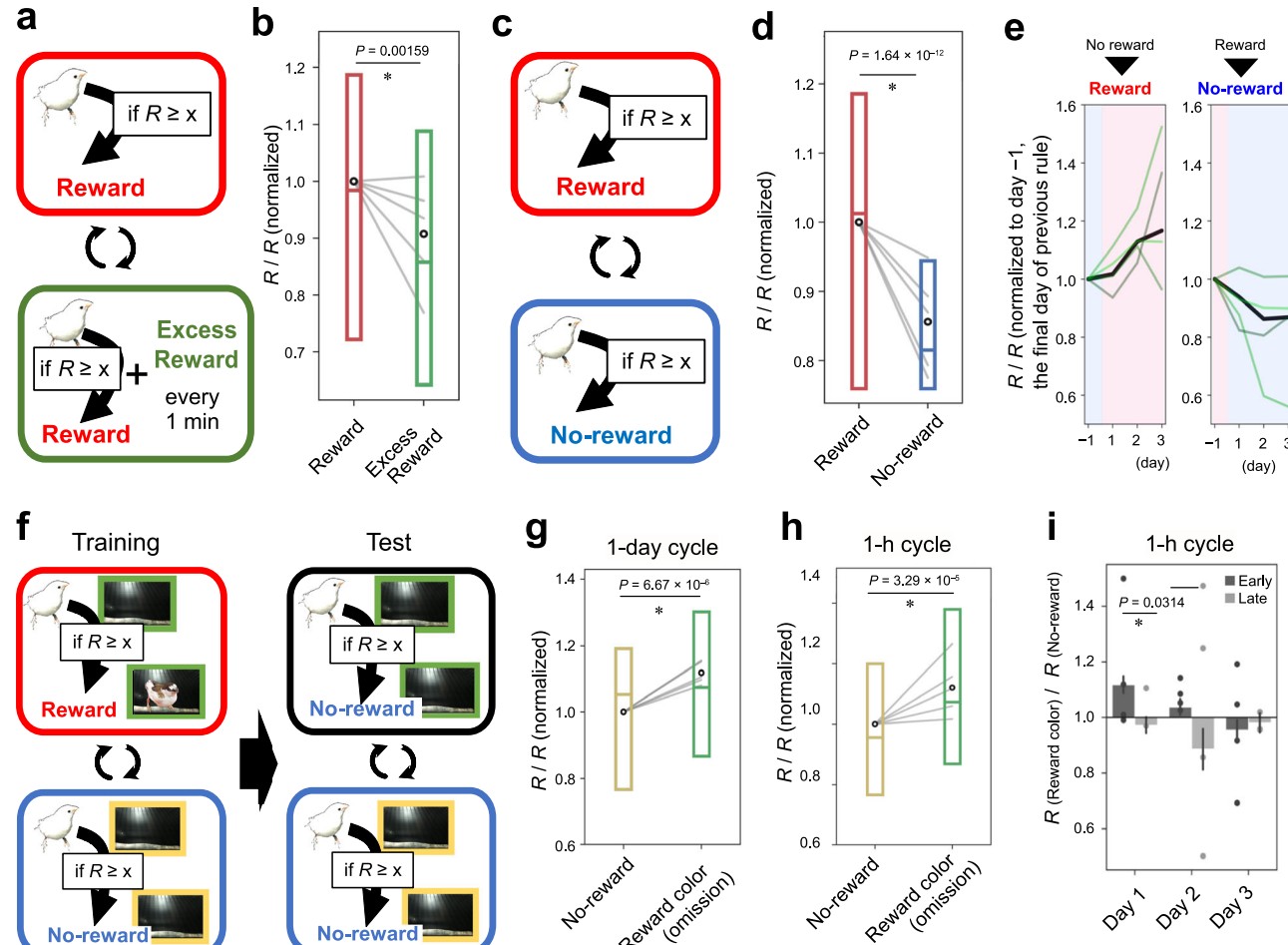

**Fig. 4 | Context-dependent modulation of song contents. a** Experimental scheme of devaluation training. "Reward" day and "Excess-reward" day were repeated each for 3 days. **b** The shift of repetition, $t(4) = 3.18$, $n = 5$ birds. **c** Experiment scheme. Reward day and "No-reward" day were repeated for 3 days. **d** The shift of repetition number, $t(4) = 7.25$, $n = 5$ birds. **e** The shift of repetition across days of "Reward" days (left, red) and "No-reward" days (right, blue) within the 3-day period. **f** Scheme of training and test sessions. Repetition training was conducted with a colored frame on the monitor. **g, h** The shift of repetition number

in the test sessions when the rule was changed for a 1-day cycle (**g**) and 1-h cycle (**h**), $t(4) = -4.55$ and $-4.21$, $n = 5$ birds. **i** Shift of the ratio of repetition number in the "Reward color" condition against "No-reward color" condition in the 1-h cycle. The ratio of early sessions (10:00–15:00) and late (17:00–22:00) sessions within the days are shown. Throughout the panels, the box plot shows the median and first and third quantiles with the mean shown as a circle, $P$ values, two-sided paired $t$-test; $t(4) = 2.16$, $1.71$, $-0.347$ from day 1 to day 3; $n = 5$ birds.

enough to surpass the threshold (Fig. 4c). To analyze whether birds can modulate their behavior depending on the context, we conducted three consecutive "Reward" days followed by three "No-reward" days. We observed that repetition of the target syllable was significantly reduced on "No-reward" days (Fig. 4d). Within the three-day periods, the repetition number increased or decreased steadily across days, with the most significant change observed on the second day of the "Reward" and "No-reward" periods (Fig. 4e).

Next, we analyzed whether songbirds are capable of changing the contents of their songs volitionally. For this purpose, we trained the birds with the "Reward-No-reward" day condition with a green- or yellow-colored frame on the LCD monitor[14] (Fig. 4f). These colors were alternatively presented according to the rule applied at the time. Contrary to the previous experiment (Fig. 4c), the subjects were cued with the rules applied at the time by the frame color. Following the conditioning phase, we initiated a test phase during which the reward was not provided, regardless of whether the repetition surpassed the threshold in the "Reward color" condition. During this test phase, we observed that birds altered the repetition according to the frame color, even in the absence of the rewards (Fig. 4g). Notably, the adjustment in the number of repetitions occurred rapidly, with

modulation of repetition evident within 10 min following a change in frame color during the one-hour cycle of rule alteration (Fig. 4h). Furthermore, we found that this reward color-specific modulation diminished within a day, in response to repeated reward omissions. The change of repetition number between reward color and no-reward color conditions decreased during the latter sessions on each day (Fig. 4i), implying that birds possess the capacity to adaptively modify the learned modulation in accordance with contextual cues. Collectively, these results suggest that the modulation of syllable sequence occurs in a context-dependent and goal-directed manner according to subjective demands.

## Neural mechanisms underlying flexible modulation of song contents

To address the neural substrate for flexible modulation of syllable sequences, we investigated the anterior forebrain pathway (AFP), a parietal-basal-ganglia circuit in songbirds known to affect the phonology of song according to the social context of songbirds[27–29]. To analyze the role of the AFP pathway in the context-dependent modulation of song context, we conditioned birds with repetitive elongation as described before (Fig. 2a), and then ablated Area X with ibotenic

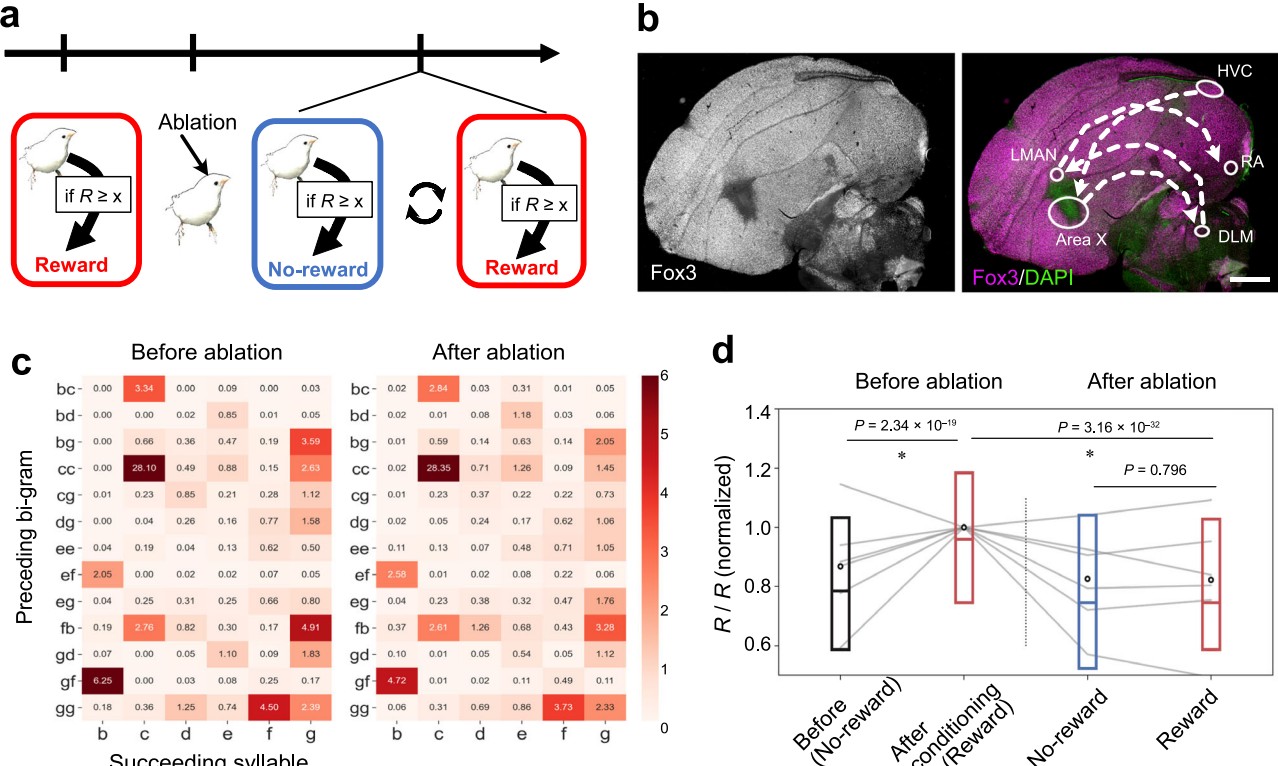

**Fig. 5 | Ablation of the brain nucleus affects the flexible modulation of songs.** **a** Overview of the experiment. **b** An example of a histological section from a bird received the nucleus ablation by ibotenic acid. The signals from fluorescent immunostaining against Fox3 (magenta), are shown with DAPI (green). A diagram of the AFP pathway is overlayed on the right. Major innervations within the nucleus are shown with dotted arrows. HVC (letter-based name); RA robust nucleus of the arcopallium, DLM medial-lateral nucleus of the dorsal thalamus, LMAN lateral magnocellular nucleus of the anterior nidopallium. Scale bar, 1 mm. **c** Tri-gram syllable transition probability matrix before and after the ablation. **d** Syllable repetition before and after the ablation. The box plot shows the median and first and third quantiles, with the mean shown as a circle. *P* values, Tukey's HSD-test, two-sided, $t(599) = 9.31$, $t(599) = 12.5$, $t(599) = 0.259$ from left to right, $n = 6$ birds.

acid (Fig. 5a, b). As previously reported[30], we observed little effect on the overall syllable transition after the ablation of Area X (Fig. 5c and Supplementary Fig. 4). The tri-gram transition analysis revealed that the transition matrices before and after Area X ablation remained highly correlated ($0.80 \pm 0.04$, Pearson's rank correlation coefficient, mean ± s.d., $n = 3$ birds; Fig. 5c), whereas unilateral ablation of HVC caused a marked reduction of the correlation ($0.45 \pm 0.24$; $n = 3$ birds; Supplementary Fig. 4), which was significantly reduced, as compared to Area X ablation (testing equality of independent of correlations, $P = 3.84 \times 10^{-7}$). After the surgery, we found that the repetition number decreased to a similar level before conditioning (Fig. 5d). Then, we further trained the birds with "Reward-No-reward" conditioning (Fig. 5a). A significant difference in repetition number was not observed between "Reward" day and "No-reward" day in Area X ablated birds (Fig. 5d), contrary to the control treated birds (Supplementary Fig. 5). We observed a significant difference in the context-dependent modulation of songs before and after the ablation ($P = 4.02 \times 10^{-7}$, LME model, $F(1199, 1199) = 25.83$). These results support the hypothesis that a specific neurocircuit is causally involved in the context-dependent flexible modulation of song contents.

## Modulation of non-repetitive locus of the songs

The Bengalese finch songs are supposed to be composed of motifs, also referred to as phrases, a stereotyped chunk of syllables that often appear together in conjunction[31,32]. Thus, to investigate the bird's ability to modulate the syntax of songs, we concentrated our analysis on the selective usage of motifs. To identify motifs in the songs of a bird, we first performed the morphological analysis of birdsongs using latticelm, a Bayesian non-parametric segmentation method used in natural language processing[33]. This segmentation identified chunks of

syllables in songs based on their pattern of actual usage (Fig. 6a). The latticelm-segmented motifs had significantly lower perplexity in predicting the upcoming syllable (Fig. 6b), demonstrating the effectiveness of this segmentation method. To further analyze the song context dependency of motif usage, we calculated the vector similarity of each motif pair by word2vec[34] (Fig. 6a). Based on this analysis, we selected the target of reinforcement to those having similar occurrences and vector similarity scores (Fig. 6c). Songs of bird #7 contained a converging locus where motifs "f-bca" and "da-bca" exclusively appeared (Fig. 6d, e). The target motif of reinforcement was set to this alternative motif pair having a high vector similarity score of +0.8 by word2vec analysis, implying that those motifs might be used in an exchangeable manner. We reinforced the subject to use "f-bca" in their songs to get the reward. After the reinforcement, a significant increase of "f-bca" occurrence over "da-bca" was observed (Fig. 6f). Then, we switched the target motif to the alternative motif "da-bca", and observed a contrary increase of "da-bca" occurrence over "f-bca" songs (Fig. 6f). This result was not caused by increasing a particular song containing "f-bca" or "da-bca," because the occurrence of "f" syllable or "da" syllable did not significantly differ among the conditions (Fig. 6g). Subsequently, we attempted to reinforce the same bird to use another pair of the motif "baca" and "dabe" having a low vector similarity score of −0.2 in word2vec analysis (Fig. 6h). The bird failed to learn to use this motif pair alternatively among the conditions (Fig. 6i). The effect of reinforcement training was significantly different on "da-bca" - "f-bca" pair to "baca"-"dabe" pair ($P = 2.43 \times 10^{-6}$, LME model, $F(199, 199) = 22.9$), indicating that not all motif transition can be trained. Similar results were obtained with the reinforcement of the divergent locus in another bird (Fig. 6j–n). Collectively, these results indicate the ability of the Bengalese finch to flexibly change the contents of the songs was

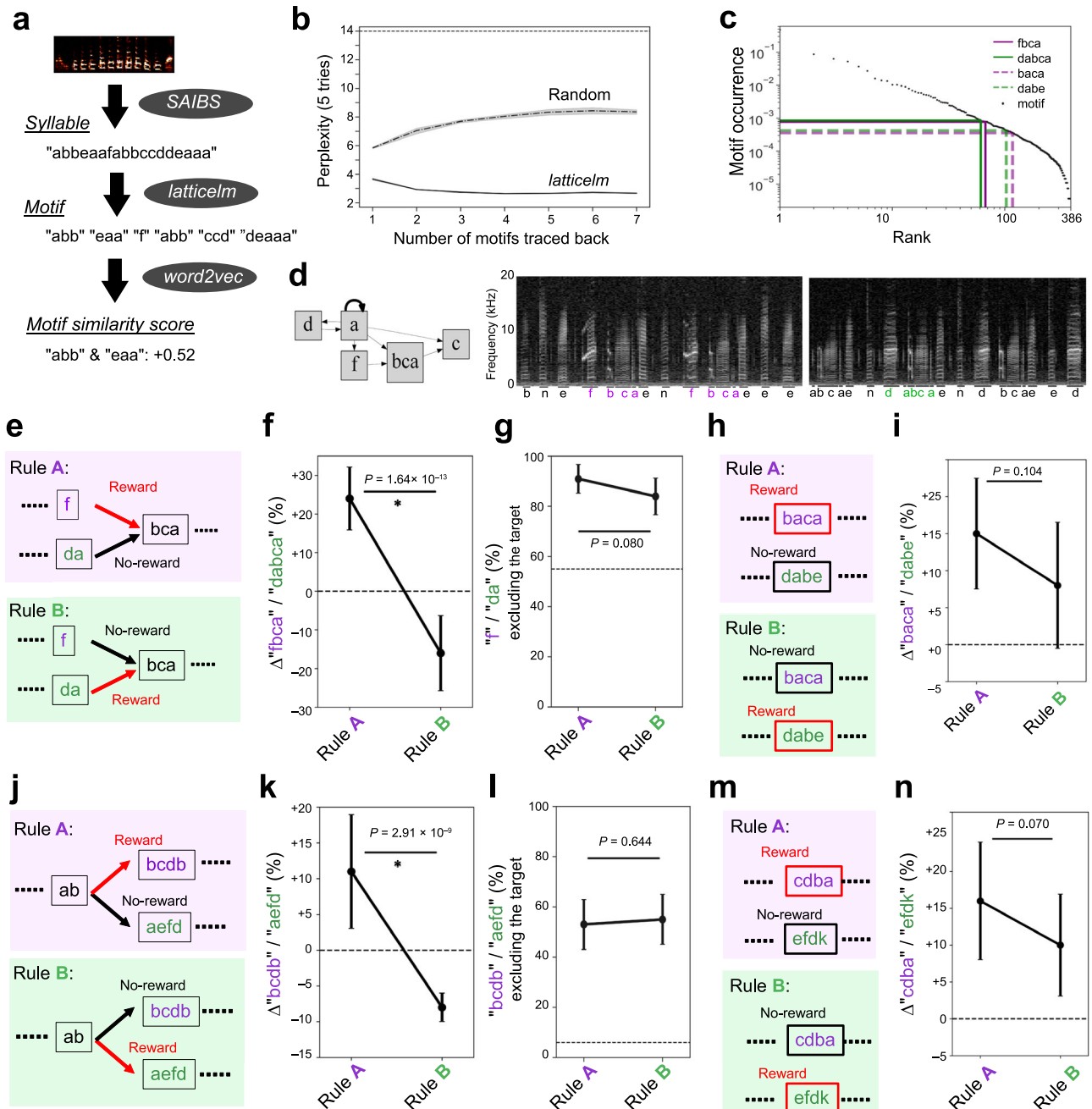

**Fig. 6 | Syntactic organization affects the flexible modulation of song contents.**
**a** Overview of the morphological analysis. **b** Perplexity results with the different number of latticelm-segmented motifs compared with randomly-segmented motifs. The perplexity in predicting the upcoming syllable of birds having a total of 14 syllables are shown. **c** Log-log plot of rank-frequency distribution of motifs. The rank of motif usage in the song corpus, and the occurrence of that motif are shown. The selected alternative targets are shown with green and purple lines (high similarity pair) and dots (low similarity pair). **d** Transition diagram and sonograms of bird #7, having "f-bca" (left) and "da-bca" (right) transition. **e** Alternative motif usage conditioning in the convergent locus. The frequency of the utterance of the target motif is shown in rules A (purple) and B (green). **f** Percent change of frequency of the alternative motif "f-bca" over "da-bca". **g** Percent change of frequency of syllable "f" and "da" excluding those used as "f-bca" and "da-bca".
**h**, **i** Conditioning of motif pair with low similarity. **j**–**n** Example of another bird that learned to use an alternative motif with a high similarity score in the divergent locus (**j**–**l**), but not with low similarity motifs (**m**, **n**). Throughout the panels, plots show mean ± s.e.m., $n = 100$ bouts; $P$ values, Tukey HSD-test, two-sided; $t(99) = 8.54$ (**f**), $t(99) = 1.77$ (**g**), $t(99) = 1.64$ (**i**), $t(99) = 6.53$ (**k**), $t(99) = 0.463$ (**l**), $t(99) = 1.83$ (**n**). The dotted lines indicate the ratios before the conditioning that were used for normalization.

not restricted to repetition; however, the locus of flexible modulation may be limited to a particular context within the songs.

## Discussion

Although previous studies have reported that songbirds employ distinct songs or calls in varying circumstances, a critical question remains regarding whether the observed changes in their behaviors stem from affective or volitional factors[16]. In the present study, we provide evidence of the Bengalese finches' ability to exert flexible control over the sequential ordering of syllables within their songs. Earlier studies have documented that birds display a range of songs from their repertoire in response to social feedback[10]. In contrast, our

findings offer a novel perspective, demonstrating that birds possess a capacity to control syllable sequence within a singular song rendition according to the social feedback and revealing the candidate circuitry responsible for this modulation.

A prior study has reported that the Bengalese finches can be trained to modify the content of their songs to avoid disturbances from a white noise feedback signal[14]. While the interpretation of whether the subjects perceive the feedback signals as positive or negative should be approached with caution[35], the resultant avoidance of using the targeted syllable suggests their procedure could be interpreted as negative reinforcement[14]. The study also demonstrated that birds can flexibly use specific syllables based on arbitrarily assigned contextual cues. Our results corroborate these observations, showing that birds can flexibly alter the sequence of syllables and further reveal that they can also modulate their songs to obtain positive rewards, such as virtual social signals. Moreover, we found that their flexible song modulations required a contingent delivery of the reward and depended on the objective value of the reward. These features suggest that these modulations occurred in a goal-oriented, volitional manner[16,36,37]. One of the distinct aspects of our study concerns the timing of feedback presentation. The prior study provided the feedback swiftly, overlayed to their vocalization, resulting in distorted auditory feedback during singing. In contrast, our experiment provided rewards only after the completion of each song bout, precluding the possibility that subjects prolong their repetition until receiving rewards, or adaptively adjust their vocalization according to the auditory errors while singing. Moreover, this feedback timing more closely resembles the natural social feedback encountered during inter-individual communication. Our results further corroborate the observation that birds can modify their song content in response to social feedback during communication[38].

Our study reveals that Bengalese finches exhibit a high degree of flexibility in controlling the length of repetition in a song bout. The observation that only the repetitive locus underwent modulation as a result of conditioning further suggests that such alteration transpires intentionally at a specific locus rather than being a byproduct of affective influence on their vocalization. Prior research has established that the repetition of syllables within a song-bout cannot be accurately modeled using a first-order Markov process, indicating that the cycle of repetition is not determined stochastically[24,25]. Furthermore, it has been revealed that some Area X projecting neurons in HVC exhibit firing patterns correlated with the numerical information pertaining to repetitive cycles[24]. These observations collectively suggest that the number of syllable repetitions within a song bout is predetermined prior to the initiation of repetition.

In the non-repetitive part of the songs, we hypothesized that learning-dependent flexible change of songs occurs at the motif level, not at the syllable level, because in many species, an action sequence consists of a chunk of behavioral elements strung together into a sequence[39]. We observed that the competency of learning-dependent modulation might differ according to the locus of the song (Fig. 6). As it was shown that birdsongs share some features with human speech concerning the hierarchical organization[40], and long-range influence on syllable sequences[41], one explanation of our failure to condition the birds to use some target motifs might be the existence of a syntactical rule of motif usage, as observed in previous behavioral studies[8,40]. However, whether there is a hierarchical mechanism underlying the sequence of motifs or whether this sequence was compositionally generated to convey different semantic information[9,42] cannot be discussed by our present research and needs to be addressed in future research with a persuading and objective methodology.

The neural mechanism that enables context-dependent change of song syntax is still unknown. In this study, we concentrated our analysis on AFP and observed its involvement in the modulation of syllable sequence depending on the context. This was in accordance with the previous observation that activity through AFP provides variation to the songs both in respect of phonology and syntax[28,29,43,44]. However, our results do not exclude other potential neural circuits of sequential modulation besides AFP pathways, such as HVC[44], Nif[45,46], MMAN[47], Uva[48], and Ad or AIV[49,50]. A yet unexplored issue is whether the context-dependent modulation of syllable orders and the spectral feature of syllables are driven by distinct or shared mechanisms. A detailed underpinning of the neural mechanism behind the flexible modulation of syllables awaits further investigation of these nuclei.

In summary, our findings demonstrated the previously unknown ability of songbirds to modulate their song contents in a goal-directed manner according to the contexts. Further assessment of the modulation mechanism will contribute to the understanding of the biological basis of flexible modulation of speech content in human language communication.

## Methods

### Animal treatment
The care and experimental manipulation of animals used in this study were reviewed and approved by the institutional animal care and use committee of the Tohoku university (2020LsA-005). All experiments and maintenance were performed following the Regulation for Animal Experiments and Related Activities at Tohoku University, and relevant guidelines and regulations[51]. Finches were purchased from Asada Chojyu and kept on a 14-h/10-h light cycle (daytime: 8:00–22:00); water and food were given ad libitum.

### Creating SAIBS program
We created a MATLAB (Mathworks) based architecture named "simultaneous automatic interpreter for birdsongs (SAIBS)". The code implementing the SAIBS and documentation are available on GitHub (https://github.com/SAIBS-paper/SAIBS). This program utilizes the following toolboxes: Audio, Deep Learning, DSP System, Image Processing, Signal Processing, Statistics, and Machine Learning.

### Song recording and training *SAIBS*
We collected the songs of 23 male Bengalese finches in this study. Songs are recorded as described before[52]. Briefly, the subject birds were isolated in a cage in a soundproof chamber (68 cm × 51 cm × 40 cm), and their songs were recorded through a microphone (ECM8000, Behringer), digitalized with a sampling rate of 44.1 kHz, 16 bit by OCTA-CAPTURE (UA-1010, Roland). Songs are extracted by SAP2011[53] and stored in the song corpus of each bird with a timestamp. Training of SAIBS was performed for each bird using the recorded song corpus. For each bird, 6000–15,000 song files were used. For syllable segmentation, we set the fast Fourier transform (FFT) window as 256 samples at 44.1 kHz, corresponding to 5.8 ms, and recorded the resultant 40-segment data, having 128 bands in the frequency axis. During this analysis, we omitted 48 frequency bands (lowest 8 and highest 40, containing mainly noise signal) to obtain audio data of 3200 dimensions. We applied a Prewitt filter to detect the edge and extract syllables, as the audio segment having vertical and horizontal edges lasting for more than 4 segments. The threshold of the filter, which was set around 0.03–0.005, was determined for each bird according to the amplitude of vocalization, background, and flapping-related noise of each recording condition. To determine whether the sequence of the extracted syllables is from songs and not from movement-related sounds or noise, we introduced a cutoff value that increases when syllables are detected while regressing according to the time-lapse of the non-syllable segment. The end of songs was defined as 2 s after the detection of the final syllable.

### Syllable detection using SAIBS
From the song corpus of each bird, we randomly selected 2000 songs from a 2-week song recording period. Each of these recorded songs

was divided into segments representing individual syllables based on continuous contours. Up to 20,000 syllables were randomly selected from this data for further analysis. The audio information of the selected syllables, with 3200 dimensions, was reduced by the t-distributed stochastic neighbor embedding (t-SNE) algorithm[17] with a perplexity parameter of 30. The syllables were then automatically clustered using density-based spatial clustering of applications with noise (DBSCAN)[18], with epsilon and minimum point parameters set at 2–2.5 and 25–30, respectively. This resulted in clustering annotated syllables into 10–20 groups, each labeled with an alphabetical letter such as "a" and "b". The parameters of DBSCAN were manually adjusted to ensure that the spectrograms of the clustered syllables were distinct from each other. We also labeled silent periods as "/" when no syllable was detected for more than 50 cells, corresponding to 290 msec. We trained a convolutional neural network (CNN), the patternnet function of MATLAB, with these clustered results as teacher signals. The dimension of the hidden layer was set from 10 to the number of extracted syllable variations that showed the minimal Akaike's information criterion (AIC). We stopped training at the epoch with the lowest cross-entropy with the validation data, where the validation error increased consecutively for six more iterations. The online detection of syllables was achieved by inputting an audio stream into the SAIBS, which underwent contiguous preprocessing through multiplication with a Hamming window function and computation with a fast Fourier transform (FFT) window of 256 samples at 44.1 kHz, corresponding to a temporal resolution of 5.8 ms. This yielded 40-segment data with 128 bands in the frequency axis, from which the lowest 8 and highest 40 frequency bands were removed. Subsequently, edge detection was applied to the remaining data in order to segment and annotate syllables using the CNN sufficiently trained offline with the subjects' songs. The entire process was completed with a latency of less than 5.8 ms, and no delay was observed in the microphone-mediated recording. The actual time lag of presenting feedback to the subjects in our experimental setup (SAIBS operating on Windows 11 (Microsoft) laptop computer of Core-i5-1135G7 CPU) was measured as $109 \pm 19$ ms after the end of songs.

### Evaluation of syllable annotation accuracy

**Similarity analysis.** To evaluate the annotation produced by various methods, we determined the sequence identity rate between the texturized sequences generated by TweetyNet[19] and those produced by each method. Since there is no existing method capable of providing an absolutely accurate annotation of birdsongs, we evaluated the relative accuracy of each methods by comparing them to the annotation generated by TweetyNet, under the assumption that the annotation by TweetyNet is accurate[19,20,54]. The sequence identity rate was computed as the ratio of matched syllables to the total number of syllables in the sequence and was shown as a percentage. We further examined any discrepancies to classify the types of errors (substitutions, insertions, and deletions) and presented the results in Fig. 1e. Additionally, we calculated the syllable match rate for each syllable by normalizing the number of correctly annotated events by the number of occurrences of that syllable in the given data. For the comparison with human annotation, the manually annotated syllable types of each researcher were compared with SAIBS annotated syllables, and their labels were converted if they had identical spectrograms. The converted results were then compared with the annotation generated by TweetyNet, which had been trained using SAIBS annotation. These calculations were performed for results obtained from individually trained SAIBS ($n = 3$, $95.28 \pm 0.30\%$) and human researchers ($n = 4$, $72.27 \pm 5.33\%$). This was statistically significant with $P = 4.84 \times 10^{-3}$ by Welch's t-test. The primary reason for the lower accuracy in human annotations is due to the inherent variability in visual inspection-based annotations. This variability is evidenced by the higher deviation observed in human annotations compared to those by SAIBS, and

significantly less coefficient variance (SAIBS 0.317%, manual 7.38%). Since our study utilizes the operant conditioning paradigm, consistent provision of the reinforcer across the songs is the major aim of annotation[26]. Thus, we consider the consistency of annotation as reflected in the coefficient variance as the indicatory of accuracy.

**Regarding reproductivity.** The calculation process of SAIBS does not involve randomness. As a result, when a recorded audio file is processed offline, it consistently produces the same annotation, i.e., yielding a sequence identity rate of 100%. However, when the annotation is applied to a stream of recorded files online, the variance in the reading frame affects the output results. To investigate this, we artificially set the reading frame to the pre-recorded audio files and processed them offline 20 times with randomized reading frames. The resulting similarity score was $98 \pm 2\%$. For this analysis, 100 song files were randomly selected from the song corpus and analyzed.

**Comparison to other annotation methods.** To compare annotations with those produced by TweetyNet, we used recorded 1000 songs from the corpus of one bird. Of those, 900 songs (totaling over 8000 s) were used to train TweetyNet using the SAIBS-generated annotation as a learning template, and an additional 100 songs (totaling ~1000 s) were used for training evaluation. The spectrogram for training TweetyNet was generated using the following parameters: fft_size, 512; step_size, 384; window_size, 106. TweetyNet was trained using following parameters: batch_size, 8; num_epochs, 8; val_step, 200; ckpt_step, 200; hidden_size, 256. To compare the annotation produced by SAIBS with manual labeling, 30 songs were annotated by four skilled researchers with at least one year of experience in manually annotating birdsong. The results of the manual annotation by each researcher were compared with the annotation created by TweetyNet.

### Operant conditioning

**Apparatus and reward.** As we used the subjects' own songs for conditioning, only adult male birds (>200 days post-hatch) were used. For operant conditioning, a subject bird was isolated in a soundproof chamber equipped with an in-panel-switching liquid crystal display (IPS-LCD, Kenowa JP-11.6-1080, resolution of 1920 × 1080, 11.6-inch screen). The monitor was located next to the cage and connected to a PC running the SAIBS program, which recorded and analyzed the song online. When SAIBS detects the utterance of the target sequence (such as repetition exceeding the threshold or the targeted motif), it outputs a movie to be displayed on the monitor. During conditioning, we presented eight movies showing a conspecific Bengalese finch framing in and out, each lasting between 7 and 32 s, with no audio. As our experimental protocol aimed to provide the subject with a natural social response as a reward, we implemented a 2 s delay in the delivery of the feedback movie in order to prevent providing an excessively swift, unphysiological reaction. The movies were presented on an image of a birdcage with metal stalks and a dark background, which was continuously displayed on the monitor throughout the experiment, even when the rewarding movie was not shown. For non-social feedback conditioning, we presented a movie wherein a circle of similar color and size to the bird shown in the original reward appears in the background.

**Conditioning of syllable repetition.** For repetition training, we only analyzed birds with syllable repetitions in their songs. We set a target syllable for repetition and a threshold number of reward presentations for each bird, resulting in a differing phonological feature of repetitive syllables for each bird. Under the "Reward" condition rule, every time *SAIBS* detected the number of repetitions of the target syllable exceeding the predetermined threshold, a movie reward was provided. For each individual bird, we established a repetition number threshold by examining their song repertoire prior to the training and

subsequently analyzing the distribution of repetition numbers. The thresholds were determined such that only a small percentage of songs exceeded the established values. The specific thresholds assigned to the six birds in our study were as follows: 4, 3, 7, 8, 11, and 7. Correspondingly, the percentages of songs surpassing these thresholds prior to the training were 3.0, 1.7, 1.5, 7.7, 1.8, and 2.9%, respectively. If the uttered song contained multiple loci of repetition, a reward was presented based on whether the number of repetitions at the locus of maximum repetition exceeded the threshold. For the analysis of repetition change, 100 songs during the period before (day −6 to −1) and after (day +3 to day +9) the start of reward conditioning were randomly selected and analyzed. For the "Not-too-much" conditioning rule, we set an additional threshold, the omission threshold, as two more syllables beyond the threshold for "Reward" conditioning. When a bird sang a song with a repetition number exceeding the omission threshold, no-reward movie was presented. Under the "Not-too-much" rule, a reward was only given if the bird sang a song with a repetition number ranging from 0 to +1 above the reward threshold. Under the "Excess-reward" conditioning rule, the reward criteria were the same as for "Reward" conditioning, but the subject was provided a reward movie once per minute regardless of their behavior. For "No-reward" conditioning, no-reward movie was provided even if the bird sang songs that exceeded the "Reward" threshold. During the "Reward" and "No-reward" conditioning, each rule was applied for three consecutive days, alternating three times. The above three conditioning rules were applied each day without informing the subjects of the rule in use. For the "Frame color" conditioning, the same procedure was followed as for "Reward" and "No-reward" conditioning, but the LCD monitor showed the image of background image or the reward movies with a green or yellow-colored frame (1.5 cm from the monitor periphery) as an arbitrary visual cue[14]. During the "No-reward" periods, the alternative color (yellow or green) was shown on the frame. A similar number of birds were assigned to yellow-frame reward conditioning and green-frame reward conditioning. After eight days of training on "Reward" day and "No-reward" day, we proceeded to the omission test trial, in which sessions showed a reward color frame but no-reward movie. This test trial was presented for one day with a reward color frame, followed by one day with a no-reward-color frame, both without showing reward movies regardless of the bird's songs. This session was repeated three times. During "Frame color" conditioning, the conditioning rules were changed at 10:00, and the songs from 11:00 to 22:00 were used for analysis. Then, we proceeded with a 1-h alternative rule presentation, in which no-reward color frame was shown for 10 min, followed by 50 min of blank, then reward color frame without reward for 10 min, followed by 50 min of a blank. This 1-h alternative rule presentation was repeated throughout the days. The songs recorded on three consecutive days were used for analysis. For each day, songs on the earlier (10:00–15:00) and latter (17:00–22:00) periods were analyzed to assess the change of songs within a day. For the analysis of songs during the 1-h alternative rule presentation, only songs uttered during the 10-min periods when the colored frame was shown were used. For analyzing the effect of the ablation of Area X, birds were first trained with "Reward" conditioning. Then, their brain was locally ablated with an injection of ibotenic acid (see the below section for surgical procedures). After the recovery from the surgery, the birds were subjected to "Reward" conditioning and "No-reward" conditioning. Each rule was applied for three consecutive days, alternatively three times. After the experiment, their ablated locus was inspected with immunostaining of the brain sections.

**Conditioning of the non-repetitive locus.** To select the target motif for conditioning, we performed the morphological analysis of birdsong to obtain "motifs," defined as probabilistically appearing chunks of elements. We do not refer to our "motifs" as "morphemes" because

we do not know if our "motifs" contain semantic information, which is often a part of definitions of "morpheme". We adopted a Bayesian approach, which assumes all unobserved features of songs as a random variable to be relevant for analyzing the syntactical aspect of birdsongs, rather than using the traditional Markov-based modeling. This study used latticelm[33], an unsupervised learning algorithm based on the Pitman-Yor language model, because this method does not utilize semantic information about the language, an aspect often included in models of natural language processing of human speech corpora. To perform morphological analysis with latticelm, we used >10,000 text-transcribed songs recorded before the start of conditioning for each bird. We set the following parameter for latticelm: n-gram, 5; annealsteps, 5; annealength, 7. We assessed the quality of morphological analysis to obtain "motifs" with a perplexity analysis, which reflects the prediction accuracy of an upcoming syllable given a certain number of morphemes before it. The range of perplexity spans from 1 to the total syllable number, where a perplexity value of 1 indicates that the upcoming syllable is entirely predictable, while a perplexity equal to the total syllable number indicates that prediction occurs at random. We performed the perplexity analysis with a bidirectional long short-term memory (BiLSTM) modeling of the "motif" orders in songs. We then used these data to train a convolutional neural network (CNN) with 100 hidden layers. To filter out non-song audio files, we only used songs consisting of more than nine "motifs". As a control, we constructed a randomized morpheme that reflects the syllable-length distribution of the motifs outputted by latticelm. We used word2vec to further analyze the resulting "motifs" by generating information vectors. The parameters we employed were the following: vector_size, 300; min_count, 100; window, 3. We calculated the vector similarity score for each pair of vectors, which reflects the syntactic relation between the two "motifs". A higher similarity score indicates that the two motifs occur in similar syntactic contexts. We used this similarity score to select the conditioning target, with a criterion of a similarity score greater than +0.8 and a frequency of utterance around 2% in the entire corpus for the high similarity pair.

**Analysis of conditioning.** For each conditioning rule, we randomly sampled 100 song files per day and recorded and analyzed the maximum repetition number within each song bout. We used a paired *t*-test to analyze the change in repetition number before and after the application of conditioning rules. The change in song structure and syllable sequence was presented as tri-gram or mono-gram transition matrices. For yielding the transition matrix, we translated the syllable sequence of all the songs during the period of interest, such as the rule application. From this data, we divided the total occurrence of each mono-gram (syllable) transition from bi-gram by the total number of transitions, resulting in a probability matrix of tri-gram transition. For simplicity of data visualization, we omitted from the matrix any transitions that occurred less than 1% in both the column and row. To analyze the significance of the change in tri-gram transition before and after the manipulations, we calculated Pearson's rank correlation coefficient. To compare these values, we performed a *Z*-transformation of the coefficient value and tested the equality of correlations.

**Surgical procedure**
Adult Bengalese finches were anesthetized with a medetomidine-midazolam-butorphanol mixture (medetomidine 30 μg/mL, midazolam 30 μg/mL, butorphanol tartrate 500 μg/mL, NaCl 118 mM; 200 μL per bird), after which ibotenic acid (0.5 μL, 2.0 mg/mL in PBS, Cayman) was injected bilaterally for Area X and unilaterally for HVC at the coordinate from the Y sinus: anterior, 5.75 mm; lateral, 1.2 mm; depth, 2.9 mm, and 2.7 mm at beak angle 60° and 70°, respectively, for Area X. Anterior, 0 mm; lateral, 2.0 mm; depth 0.75 mm at beak angle 40° was used for HVC. To inspect the ablated area, the birds

were euthanized after experiments and perfused with PBS, followed by 4% paraformaldehyde (PFA, Nakarai-tesque) in PBS. The cryosections were immunostained with Fox3 (NeuN; Millipore, MAB377; 1:750) together with 4′,6-diamidino-2-phenylindole (DAPI, Dojido, 1 µg/mL) staining. Images were taken by BZ-9000 (Keyence) using a 10 × lens, and the whole section merged images were used for the inspection of the lesioned area. The ablated area was measured by visual inspection of the stained images using Image J (US National Institutes of Health). The ablated area were 0.841, 1.28, 0.515, 0.804, 0.787, and 0.786 mm$^2$ per hemisphere for each bird.

## Statistics

The alpha score of 0.05 was used to reject the null hypothesis. The sample size was determined based on previous studies. Welch's $t$-test and paired $t$-test were used for the comparison of two data sets, and Tukey HSD-test was employed for multiple comparisons. LME model was used to compare the effects of manipulation on multiple experiments.

## Reporting summary

Further information on research design is available in the Nature Portfolio Reporting Summary linked to this article.

## Data availability

Source data is provided as a source data file. All data underlying the presented result are contained in this file. Source data are provided with this paper.

## Code availability

The code used in this study are available on GitHub: https://github.com/SAIBS-paper/SAIBS which is archived in Zenodo with the identifier [https://doi.org/10.5281/zenodo.10802632][55].

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

## Acknowledgements
We thank K. Inui, Y. Ogawa, K. Hamaguchi, H. Fujimoto, S. Aoki, and the members of Abe Laboratory at Tohoku University for their help and fruitful suggestions. This study was funded by JSPS KAKENHI JP23K18252, JP22H05482, JP21K19424, JP21H05608, JP19H04893, JP19H03319 (to K.A.), JP22KJ0225 (to T.K.), and Tohoku University Research Program "Frontier Research in Duo" No. 2101 (to K.A.).

## Author contributions
Conceptualization: K.A., T.K. Methodology: T.K. Investigation: T.K., M.F. Visualization: T.K. Funding acquisition: K.A. Project administration: K.A. Writing—original draft: K.A., T.K. Writing—review and editing: K.A., T.K.

## Competing interests
The authors declare no competing interests.
