## [Peer Review File · Nature Communications]

Goal-directed and flexible modulation of syllable sequence within birdsongREVIEWER COMMENTS

Reviewer #1 (Remarks to the Author):

In this study, the authors investigated the ability of a songbird species, the Bengalese finch, to adjust syllable sequences. To do so, they first introduce a new method for online automatic identification of individual song syllables. The accuracy of this method, named SAIBS, is compared to a previous method. The authors then succeeded in using the SAIBS method together with a video cue in order to train birds to sing either more repetitions of a specific song syllable, a specific number of repetitions or even to sing more frequently some sequences of syllables. A lesion of the avian basal ganglia (area X) prevent the birds to modulate the number of repetitions of a specific song syllable, thus highlighting the contribution of that nucleus in the control of the syllable sequencing.

This paper provides new insights on the control of various type of syllable sequencing (syllable repetitions and syllable ordering). It extends previous report on syllable sequencing in Bengalese finches (e.g. Veit et al, eLife, 2021).

Data are overall well presented and the key claims of the manuscript are well supported by the data. I have spotted some mistakes/elements that need clarification (see detailed comments below).

Methods are overall well described but some more details are required to ensure that this work can be reproduced (see comments below).

There are some typos and what appear as English spelling errors (I have noted some below).

Please do process to a proper check of English grammar and spelling throughout the manuscript, figures and legends.

Detailed comments

Rapid, online decoding of syllable sequence in birdsongs

L.54-56: I guess SAIBS corresponds to the decoder, not the quasi-real-time manner. Please rephrase the following sentence: For this purpose, we developed a song decoder that operates in a quasi-real-time manner called simultaneous automatic interpreter for birdsongs (SAIBS).

Figure 1: there are no labels for the x and y axes on the spectrograms in 1b and 1c. Error bars are cropped on fig 1e, please change the scale of the y axis so we can also see insertions and deletions.

Syllable annotation results: there are no information provided in this section (l.71-89) on the number and type of birds used for estimating the accuracy and reproducibility of SAIBS. I guess it is based on a single adult Bengalese finch? Also, how many songs, syllables were used for training and for testing?

The authors claim that "SAIBS was significantly more accurate than manual annotation and comparable to cutting-edge song decoders such as TweetyNet" (L.72-73), but I cannot see in the main text any statistical test confirming it. The authors provide info on the sequence identity rate of syllable annotations.

L80-81: "The SAIBS decoder was significantly more accurate than the manual annotation (four skilled researchers; $72.27 \pm 5.33\%$, Welch's t-test, $P = 4.84 \times 10^{-3}$; Supplementary Fig. 1a)" -> the suppl. Fig1a shows the comparison between TweetyNet and human annotation, not between SAIBS and human annotation. Also, how did the authors process the manual annotations from the four skilled researchers and perform the Welch's t-tests to compare the human annotations to the SAIBS decoder? Did they run 4 Welch's t-tests? Or did they average the classification performed by the 4 researchers? Some more information is required.

Fig1d : please provide scale bar for the heatmap

Operant conditioning of syllable repetition in songs

L.102-103: it is more a matter of definition, but to my point of view, the use of the term "aversive" is inappropriate: the aversive valence of the auditory stimuli (such as a white noise) can be only deduced by the behaviour of the bird exposed to such stimuli. One can imagine that some birds might consider a white noise as a positive reinforcer. It has been reported in a similar (pitch-shifting) paradigm that a visual stimulus (transient light off) can act as a positive or a negative reinforcer according to the bird status (Zai et al, Nat. Comm 2020). I would thus just remove the

term "aversive" in the main text.

L.106: the authors mention a pilot study that allowed them to assess that the short movies of conspecifics elicited social behaviour. But there are no further details on that pilot study. What type of social behaviors were elicited? Were these behaviors only elicited by movies of conspecifics compared to other types of videos? Also, there are not many details provided in the main text and methods on the content of the short movies that were presented. What was the kind of behaviour of the conspecific in the short movie? Was the bird eating, sitting, moving around, etc? Was it a female conspecific? It would be nice to have access to one example video as a supplementary file. To my point of view, these are important details to provide to fully consider that the short movies of the conspecific may act as a social cue. Also, did the authors perform a control experiment in which something else than a conspecific (e.g. a red dot) was projected on the image of the cage contingently to the syllable repetition if $R \geq x$? This would be an important point to justify that the movie of the conspecific is a social cue and not just an informative cue that the subject uses. In other words, without the testing of a control video, I am not sure that the authors can be 100% sure that the video is a "social" cue. I am referring again to Zai et al, Nat Comm 2020 who showed that zebra finches can adjust their song to flash of light. It might thus be that in the present study, Bengalese finches adjust their song because of the contingent visual feedback (which is already great), but independently of its putative social value.

Fig2e & h : please provide scale bar for the heatmaps (same comment for fig 3b, 3d, 5c)

L.112-113: typo: repetitive syllables

Fig2c & d & L.117: how many days of experiments separate before and after conditioning ?

Song change occurs at a defined locus within the song

Did the authors also looked at the change in the average number of loci per song bout? I am indeed wondering whether the birds in which the modulation was not observed in the first locus tend in general to produce song bouts that includes more loci.

L.143 & Fig 3b : "the probability of target syllable "c" after "c-c" was increased by 8.09%" : a difference of percentages is not equivalent to a percentage of differences. The difference between 26.93 and 35.02 is 8.09 percentage points. Same comment for the mean change of non-target syllable transition (0.33 percentage points). The y label on fig 3b should not be "Preceding bigram" (same comment for fig5c)? Also, in the legend, "repletion": do the authors mean repetition? L158: although the legend of fig4a provides the info on the number of days (3) in each condition "reward" days vs "excess-reward" days, it is important to also provide that information in the main text.

Neural mechanisms underlying flexible modulation of song contents

Please provide the number of animals with HVC lesion. For how long the birds were trained post-lesion?

Modulation of non-repetitive locus of the songs

Overall, although I am not familiar with the algorithms used by the authors (latticeLM and word2vec), which makes it quite difficult to assess the validity of the method, it does seem like a promising approach. Yet, some more details should be provided.

For example, about the "Perplexity" index: The authors provide the following definition: it "reflects the prediction accuracy of an upcoming syllable given a certain number of morphemes before it" (L.594-595). It would be interesting for the reader to get some more information such as 1/ the range of the perplexity index ([min max]), 2/ what mean high and low values (I guess the lower it is, the higher is the prediction accuracy). On Fig6b, there is a dotted line at the y value 14. Does it reflect the max of the perplexity index?

Fig6c: I am having hard times in understanding what is being plotted here. Please provide some more information. E.g. is it all the different motifs identified by latticeLM that a single bird produces? How is the motif occurrence computed?

It is important and great that the authors provide the detailed values of the settings of latticeLM (n-grams, annealsteps, annealength) and word2vec (vector_size, min_count, window) they have used. But how these settings were decided? How does it change the results?

L204: typo: contrary to

Fig6a: typo: motif similarity score

Methods

L.451: typo: algorithm

L.497: typo: reading frame

L.507-510: "To compare the annotation produced by SAIBS with manual labeling, 30 songs were annotated by SAIBS by four skilled researchers with at least one year of experience in manually annotating birdsong". I do not understand the section "30 songs were annotated by SAIBS by four skilled researchers": the songs were annotated by SAIBS and by 4 researchers? Also, L510-511: if the authors are comparing SAIBS with manual annotation, why is the comparison made between manual annotation and TweetyNet?

Throughout the manuscript, statistical tests value and degrees of freedom should be provided.

Reviewer #2 (Remarks to the Author):

This study is an excellent complement to the recent paper by L. Veit and the Brainard lab (citation #14). Both of these studies show that songbirds can volitionally modulate song structure in a context-dependent way, driven by reward. Notably, both studies show that the change in song syntax can be cued with visual stimuli.

The present work includes a number of novel contributions including the real-time decoding of song structure using the SAIBS software, and the delivery of reward that comes after the song, in the form of a video playback of another bird. This could relate to the "social reinforcement" described in West and King Nature, 1988, which indicated that females signal to males which songs are most attractive with a wing-flip.

Notably, this work includes a lesion study that indicates the involvement of the basal ganglia area X in this form of learning.

My only suggestion for improvement of the manuscript is to include a more thorough discussion of how this work relates to citation #14, L. Veit's paper. The introduction and concluding sentence of the discussion claims novelty for the observation of songbirds volitionally modulating song content according to different contexts. My understanding is that this ability was already dramatically demonstrated in Veit et al. 2021. It would be great if the discussion and introduction could address the similarities and differences of the two studies more thoroughly. For example, the nature of the reward signal is distinct.

I find this paper a valuable contribution to the field and even though there is a lot of overlap with Veit 2021, the details of the two studies are different, particularly in the nature of the reward signal, and this new paper stands on its own as a significant novel contribution that provides further information about about the nature of real-time, reward-driven, context-dependent control of song syntax.

Reviewer #3 (Remarks to the Author):

In this paper, the authors present a set of results relating to the ability of adult Bengalese finches to modulate the sequential structure of their songs. They first describe an online algorithm that annotates syllables as they are produced by a singer in quasi real-time. They then use this annotation method to track the sequential structure of song as it's produced in an appetitive operant training procedure. Specifically, access to short videos of conspecifics is made contingent on the repetition of a target syllable a certain number of times. This training leads to an increase in the number of target syllable repetitions, and other operant manipulations (reward devaluation and extinction) modulate repetition rates in expected ways according to standard operant learning theory. When different reward contexts are cued visually, across either days or 1-hour blocks, repetition rates vary according to the cue (even over the short term in the absence of reward). The authors then report that lesions to the anterior forebrain pathway (Area X) reduce the learned syllable repetition rates and block subsequent context-dependent repetition changes. Finally, the authors identify target (non-repeat) syllable transitions and show that operant reinforcement can be used to modulate the frequency of these transitions as well.

Overall, there is a great deal to like in this paper. After reading it closely, I find myself generally sympathetic to the main conclusion that Bengalese finches can voluntarily modulate the sequential structure of their songs (at least to a limited degree). However, the paper is not well-written, the data are rather sloppily presented, and some of the conclusions made are not justified or are irrelevant. Thus, there are a number of serious shortcomings that preclude publication, at least in its present form. I detail the most significant of my concerns below:

Main effect of operant training; Figure 2:

- The various dependent measures plotted in panels c, d, e, g, h are not defined. One can intuit that the authors are trying to show changes in the number of repetitions in the operantly targeted syllable, but why is the same behavioral effect captured with five different measures that change (for no apparent reason) across the panels. This does not instill confidence in the stability of the results.
- A paired -t-test is likely inappropriate for the data shown in Fig2C, as there is one bird that is clearly an outlier and differences may not be normally distributed. Similarly if we are supposed to compare the effect in 2c to the lack of an effect in 2d, then an LME model should be used.
- Why are there data for 9 birds shown 2c and only 6 birds in 2e?
- This looks like classic operant shaping. They should make a note of this in the discussion.

Relating to the main effects in figure 2: What exactly is the main hypothesis? Is it that any change in sequential structure that can be modified by operant learning is grounds for flexible control? Or are there only some forms of operant learning induced sequential changes that justify this claim. As it is the authors present several results showing various putative changes, some of which are stronger than others, and we are given no guide for how to interpret these in the context of the primary or secondary hypotheses, because no actual hypotheses are ever stated. The paper would benefit tremendously from a clear articulation of why particular manipulations and analyses are conducted.

The authors show no significant difference in the number of supra-threshold repetitions ($x+1 < R$) between "reward" and "not-too-much" reinforcement conditions (fig 2g, panel 3). This argues against the stated conclusion that birds could "... maintain them [syllable repetitions] within a specific range", and instead supports the conclusion that the upper bound on the range is constrained by non-operant factors.

It is difficult to interpret the results regarding "loci of modulation" without further experimentation. If the authors wish to claim that the locus depends on the "strategy" of each bird, then they should devise an empirical test (presumably involving selection of different targets within birds) to show that a proposed strategy is better than chance at explaining why a specific bird modulates the repetition at a specific locus. If a specific strategy cannot be identified, it is unclear what this section adds to the overall paper.

The overall statistical approach is weak:

I've pointed out a places where choice to use a paired t-test is not appropriate, but similar arguments might be applied to several other analyses, where authors treat select comparison as independent even when they involve the same subjects under different conditions. For example, in figure 6, panels f-g are not independent. Rather than looking at two t-tests, an LME should be used to directly test the effect of the training. (Same for panels k-I). Likewise, when discussion the lesion effects, the author's claims rest on within subjects comparisons across learning conditions (Fig 5d), but comparison to the control animals (shown only in supplementary Fig 4) is qualitative -i.e. there is an effect in one group but not the other. A LME model should be used to directly, and quantitatively compare the lesions and control birds.

The extent of the lesion in each animal is not quantified, but needs to be.

Discussion:

The authors need to do more to differentiate their results from the existing literature which (as they correctly note) contains a large number of cases where aversive auditory feedback has been used to alter the singing behavior of different birds. Although the majority of these studies have

focused on reward dependent modulation of spectral characteristics of song, it is not immediately clear that the kinds of effects observed here are necessarily distinct. One conservative interpretation is that the reported effects are just a versions of the reward mediated vocal learning that has already been widely reported.

I would suggest dropping the "volitional" term from the title and the body of the paper. The strong assumption here is that an operantly trained behavior is equivalent to a voluntary one, but this is (and has been) a contentious point in the animal behavior literature for some time. Consider, for example, the case of addiction in which behaviors are acquired operantly, but which become involuntary. I am not sure that the present work supports any progress in this age old debate, and I don't think that there is much to be gained by forcing this kind of wording on the reader.

RESPONSE TO REVIEWERS' COMMENTS

Reviewer #1 (Remarks to the Author):

In this study, the authors investigated the ability of a songbird species, the Bengalese finch, to adjust syllable sequences. To do so, they first introduce a new method for online automatic identification of individual song syllables. The accuracy of this method, named SAIBS, is compared to a previous method. The authors then succeeded in using the SAIBS method together with a video cue in order to train birds to sing either more repetitions of a specific song syllable, a specific number of repetitions or even to sing more frequently some sequences of syllables. A lesion of the avian basal ganglia (area X) prevent the birds to modulate the number of repetitions of a specific song syllable, thus highlighting the contribution of that nucleus in the control of the syllable sequencing.

This paper provides new insights on the control of various type of syllable sequencing (syllable repetitions and syllable ordering). It extends previous report on syllable sequencing in Bengalese finches (e.g. Veit et al, eLife, 2021).

Data are overall well presented and the key claims of the manuscript are well supported by the data. I have spotted some mistakes/elements that need clarification (see detailed comments below).

Methods are overall well described but some more details are required to ensure that this work can be reproduced (see comments below).

There are some typos and what appear as English spelling errors (I have noted some below). Please do process to a proper check of English grammar and spelling throughout the manuscript, figures and legends.

Detailed comments

Rapid, online decoding of syllable sequence in birdsongs

L.54-56: I guess SAIBS corresponds to the decoder, not the quasi-real-time manner.

Please rephrase the following sentence: For this purpose, we developed a song decoder that operates in a quasi-real-time manner called simultaneous automatic interpreter for birdsongs (SAIBS).

Reply: Thank you for pointing out the error. We have corrected the sentence as follows:

“For this purpose, we developed a song decoder called SAIBS, which can annotate the syllable orders with in songs in quasi-real-time.” (line: 53-54)

Figure 1: there are no labels for the x and y axes on the spectrograms in 1b and 1c. Error bars are cropped on fig 1e, please change the scale of the y axis so we can also see insertions and deletions.

Reply: We have included labels for x and y axes on the spectrograms on Fig 1b, 1c and Supplementary Fig. 2b. Error bar on Fig. 1e was not cropped because the upper limit of percentage is 100%. We correct the visual appearance of error bars to show “T” shape. We adjusted the scale of y axis to magnify the insert and deletion.

Syllable annotation results: there are no information provided in this section (l.71-89) on the number and type of birds used for estimating the accuracy and reproducibility of SAIBS. I guess it is based on a single adult Bengalese finch? Also, how many songs, syllables were used for training and for testing?

Reply: For the analysis, 100 song files were randomly selected and used for analysis, as previously mentioned in the method section. We have now also added this description to appropriate location (*line: 623-624*). Figure 1 shows the result from a single Bengalese finch. In the revised manuscript, we have included the results from another Bengalese finch alongside those of zebra finches in Supplementary Figure 2. The number of song used for analysis shown in Supplementary Figures 1 and 2 has been indicated in their respective legends.

The authors claim that “SAIBS was significantly more accurate than manual annotation and comparable to cutting-edge song decoders such as TweetyNet” (L.72-73), but I cannot see in the main text any statistical test confirming it. The authors provide info on the sequence identity rate of syllable annotations. L80-81: “The SAIBS decoder was significantly more accurate than the manual annotation (four skilled researchers; $72.27 \pm 5.33\%$, Welch’s t-test, $P = 4.84 \times 10^{-3}$; Supplementary Fig. 1a)” -> the suppl. Fig1a shows the comparison between TweetyNet and human annotation, not between SAIBS and human annotation. Also, how did the authors process the manual annotations from the four skilled researchers and perform the Welch’s t-tests to compare the human annotations to the SAIBS decoder? Did they run 4 Welch’s t-tests? Or did they average the classification performed by the 4 researchers? Some more information is required.

Reply: Technically, a direct comparison of manual labeling versus SAIBS labeling is meaningless, so we avoid describing the accuracy score of such comparison. This is because no method can obtain the truly correct answer of birdsong annotation, so each comparison have to be relative. For quantitative and statistically compare the two methods, we hypothetically set the correct answer to be the one created using the third party created TweetyNet. Since TweetyNet requires annotated template for learning, we used the annotation created automatically by SAIBS. We consider this “TweetyNet trained with SAIBS created annotation” as the ground truth for comparison, and compared the accuracy of “TweetyNet trained with SAIBS created annotation” versus manual annotation (Supplementary Fig. 1a) and “TweetyNet trained with SAIBS created annotation” versus SAIBS online decoder (Supplementary Fig 1b). The sequence identity rate for each researcher was compared to TweetyNet. The result from four researcher were $72.27 \pm 5.33\%$. Then we performed the same process with three independently trained SAIBS and got the score of $95.28 \pm 0.30\%$. This was statistically

significant with $P = 4.84 \times 10^{-3}$ by Welch's t-test. We add some description to explain the rationale for such comparison in the methods section. (line: 529-532)

Fig1d : please provide scale bar for the heatmap

Reply: We have included the scale bars for all heatmap figures (Fig. 1d, 2f, 3b, 3d, 5c, and Supplementary Fig 1ab, 2ac).

Operant conditioning of syllable repetition in songs

L.102-103: it is more a matter of definition, but to my point of view, the use of the term "aversive" is inappropriate: the aversive valence of the auditory stimuli (such as a white noise) can be only deduced by the behaviour of the bird exposed to such stimuli. One can imagine that some birds might consider a white noise as a positive reinforcer. It has been reported in a similar (pitch-shifting) paradigm that a visual stimulus (transient light off) can act as a positive or a negative reinforcer according to the bird status (Zai et al, Nat. Comm 2020). I would thus just remove the term "aversive" in the main text.

Reply: The term "aversive" was used by the authors to describe their results in previous studies by Tumer (2007) and Veit et al., (2021). In this paper they referred to their experimental paradigm as "aversive reinforcement training" and consider the white noise used in their experiment as "aversive white noise". In the initial manuscript, we adopted their terminology; however, following the comments from the reviewers, we significantly toned down our statements by omitting the term "aversive" from the main text where it was inappropriate. Additionally, we included a statement in the discussion section briefly addressing the caution required when interpreting the valence of stimuli. (line: 268-271)

L.106: the authors mention a pilot study that allowed them to assess that the short movies of conspecifics elicited social behaviour. But there are no further details on that pilot study. What type of social behaviors were elicited? Were these behaviors only elicited by movies of conspecifics compared to other types of videos? Also, there are not many details provided in the main text and methods on the content of the short movies that were presented. What was the kind of behaviour of the conspecific in the short movie? Was the bird eating, sitting, moving around, etc? Was it a female conspecific? It would be nice to have access to one example video as a supplementary file.

Reply: In the revised manuscript, we included the movie as Supplementary Movie 1. We also included a brief description about the typical behaviors we observed by movie presentation. (line: 106-107).

To my point of view, these are important details to provide to fully consider that the short movies of the conspecific may act as a social cue. Also, did the authors perform a control experiment in which something else than a conspecific (e.g. a red dot) was projected on the image of the cage contingently to the syllable repetition if $R \geq x$? This would be an important point to justify that the movie of the conspecific is a social cue and not just an informative cue that the subject uses. In other words, without the testing of a control video, I am not sure that the authors can be 100% sure that the video is a "social" cue. I am referring again to Zai et al, Nat Comm 2020 who showed that zebra finches can adjust their song to flash of light. It might thus be that in the present study, Bengalese finches adjust their song because of the contingent visual feedback (which is already great), but independently of its putative social value.

Reply: As suggested by the reviewer, we conducted a control experiment wherein a movie with non-social contents was provided contingently upon repetition that surpass the threshold. This non-social movie has been included as Supplementary Movie 2. Unlike the results observed with reinforcement using our original social movie, there was no noticeable change in repetition in this case. These results are now depicted in Fig. 1e. This experiment further strengthened our conclusions that social signals are more effective in motivating the song change compared to the simple contingent presentation of non-social signals.

Fig2e & h : please provide scale bar for the heatmaps (same comment for fig 3b, 3d, 5c)

L.112-113: typo: repetitive syllables

Fig2c & d & L.117: how many days of experiments separate before and after conditioning ?

Reply: We have included scale bars for heatmaps, and corrected the typo. Days analyzed for Fig 2 was derived from day -6 to -1 versus day 3 to day 9. Following description is now added to the methods section.

“For the analysis of repetition change, 100 song during the period of before (day -6 to -1) and after (day +3 to day +9) the start of reward conditioning was randomly selected and analyzed.” *(line: 598-600)*

Song change occurs at a defined locus within the song

Did the authors also looked at the change in the average number of loci per song bout? I am indeed wondering whether the birds in which the modulation was not observed in the first locus tend in general to produce song bouts that includes more loci.

Reply: Our procedure provide reward only when the repetition in any single locus of repetition exceeds the predetermined threshold. Thus, the total number of repetitive syllable in a song bout is not directly related to the reward. According to the reviewer's comment, we analyzed the change in the average number of loci per song bout before and after the training. We found no significant change in the number of loci (result shown in the right, $P = 0.21$ paired t-test, $n = 9$ birds). We further analyzed whether the birds in which the change of repetition in the first locus (bird#1,3,5) extends the number of loci, and found no significant effect either.

L.143 & Fig 3b : “the probability of target syllable "c" after "c-c" was increased by 8.09%” : a difference of percentages is not equivalent to a percentage of differences. The difference between 26.93 and 35.02 is 8.09 percentage points. Same comment for the mean change of non-target syllable transition (0.33 percentage points). The y label on fig 3b should not be “Preceding bi-gram” (same comment for fig5c)? Also, in the legend, “repletion” : do the authors mean repetition?

Reply: The percentage shown on Fig 3 shows the transition probability of syllable transitions. According to the reviewer's suggestion, we have now included the fold change difference before and after the training: 8.09% equals to 1.3 fold change vs 0.33% equals to 0.90 fold change (*line: 153-155*). However, we have chosen to retain the percentage description, as we believe illustrating the percentage is more important since this values reflects the magnitude of change within the overall tendency of syllable usage. This is because the fold change difference can be significantly affected by rare transitions. For a brief example, if a transition appeared only once (for example, 0.1%) before training appeared twice after training (0.2%), it represent 0.1% increment on percentage but accounts 2.0 fold increase in the fold change presentation. While such drastic distortion was mitigated by selecting the top 2-30 transitions, but the relatively rare transition still skew the values. Thus, we include both the percentage difference and fold change difference of each syllable transition.

We have corrected the typo-error in the figure and legend. “Proceeding bi-gram” to “Preceding bi-gram”, “repletion” to “repetition”.

L158: although the legend of fig4a provides the info on the number of days (3) in each condition “reward” days vs “excess-reward” days, it is important to also provide that information in the main text.

Reply: We have included the days we analyzed for “excess-reward” conditioning in the main text (*line: 171*).

Neural mechanisms underlying flexible modulation of song contents

Please provide the number of animals with HVC lesion. For how long the birds were

trained post-lesion?

Reply: The number of birds with HVC lesion was three. These are now included in the revised text (*line: 214*). In addition, we noticed that the description of HVC lesion was not correctly described. The lesion on HVC was executed unilaterally because bilateral lesion to HVC abolish singing. This was corrected in the revised manuscript (*line: 213, 686*). Training or assaying of post surgery (ibotenic acid or vehicle) birds were conducted three consecutive days, alternatively for three times. This was already described in the methods section (*line: 632-634*).

Modulation of non-repetitive locus of the songs

Overall, although I am not familiar with the algorithms used by the authors (latticekm and word2vec), which makes it quite difficult to assess the validity of the method, it does seem like a promising approach. Yet, some more details should be provided.

For example, about the “Perplexity” index: The authors provide the following definition: it “reflects the prediction accuracy of an upcoming syllable given a certain number of morphemes before it” (L.594-595). It would be interesting for the reader to get some more information such as 1/ the range of the perplexity index ([min max]), 2/ what mean high and low values (I guess the lower it is, the higher is the prediction accuracy). On Fig6b, there is a dotted line at the y value 14. Does it reflect the max of the perplexity index?

Reply: Dotted line at 14 indicate the maximum syllable number of this subject. The range of perplexity spans from 1 to the total syllable number, where a perplexity value of 1 indicates that the upcoming syllable is entirely predictable, while a perplexity equal to the total syllable number indicates that prediction occurs at random. This explanation is now included in the methods section (*line: 652-655*). We also include more description in the Figure legend to aid understanding of Fig 6b and 6c.

Fig6c: I am having hard times in understanding what is being plotted here. Please provide some more information. E.g. is it all the different motifs identified by latticekm that a single bird produces? How is the motif occurrence computed?

Reply: We have included a more detailed description regarding the content plotted in this graph within the Figure legends (Fig. 6). Log-log plot of rank-frequency distribution is a common approach utilized not only in the field of natural language processing and linguistics, but also across a variety of domains such as frequency distribution of gene expression within cells (Corral et al., *PLoS ONE* 2015; Wang G, *PLoS ONE* 2020). It is empirically known that the frequency of the appearance of an element, like words in human language, is inversely proportional to their rank of usage, a principle commonly referred to as “Zipf’s law”. However, as this is an empirical law,

and whether it is truly applicable to the motif usage of songbirds extends beyond the focus of this study, we have chosen not to include such statements.

It is important and great that the authors provide the detailed values of the settings of latticelm (n-grams, annealsteps, annealength) and word2vec (vector_size, min_count, window) they have used. But how these settings were decided? How does it change the results?

Reply: We set the parameters based on the recommended parameters for usage in human language analysis. The same parameter settings are used for the analysis of songs from each bird.

L204: typo: contrary to

Fig6a: typo: motif similarity score

Methods

L.451: typo: algorithm

L.497: typo: reading frame

Reply: We corrected all the typos pointed by the reviewer.

L.507-510: “To compare the annotation produced by SAIBS with manual labeling, 30 songs were annotated by SAIBS by four skilled researchers with at least one year of experience in manually annotating birdsong”. I do not understand the section “30 songs were annotated by SAIBS by four skilled researchers”: the songs were annotated by SAIBS and by 4 researchers? Also, L510-511: if the authors are comparing SAIBS with manual annotation, why is the comparison made between manual annotation and TweetyNet?

Reply: We are sorry for incorrectness in the initial description. The correct one is : “To compare the annotation produced by SAIBS with manual labeling, 30 songs were annotated by ~~SAIBS by~~ four skilled researchers with at least one year of experience in manually annotating birdsongs.” (line: 560-564)

Partly due to our mis-description in the original manuscript, we believe there is a misunderstanding in the reviewer about the method to compare the accuracy of annotations between two methods (manual versus SAIBS online decoder)

Direct comparison of SAIBS vs manual labeling is impossible. This is because no method can obtain the truly correct answer of birdsong annotation. For executing a quantitative and statistical comparison of two methods, we hypothetically set the correct answer to be the one created using the third party created TweetyNet. Since TweetyNet requires annotated template for learning, we used the annotation created automatically

by SAIBS. We consider this “TweetyNet trained with SAIBS created annotation” as the ground truth for comparison, and compared the accuracy of “TweetyNet trained with SAIBS created annotation” versus manual annotation (Supplementary Fig1. a) and “TweetyNet trained with SAIBS created annotation” versus SAIBS online decoder (Supplementary Fig. 1b). the result was $72.27 \pm 5.33\%$ versus $95.28 \pm 0.30\%$ with statistical significance with $P = 4.84 \times 10^{-3}$ by welch’s t-test.

Throughout the manuscript, statistical tests value and degrees of freedom should be provided.

Reply: We have now included statistical values and degrees of freedom where necessary.

Reviewer #2 (Remarks to the Author):

This study is an excellent complement to the recent paper by L. Veit and the Brainard lab (citation #14). Both of these studies show that songbirds can volitionally modulate song structure in a context-dependent way, driven by reward. Notably, both studies show that the change in song syntax can be cued with visual stimuli.

The present work includes a number of novel contributions including the real-time decoding of song structure using the SAIBS software, and the delivery of reward that comes after the song, in the form of a video playback of another bird. This could relate to the "social reinforcement" described in West and King Nature, 1988, which indicated that females signal to males which songs are most attractive with a wing-flip.

Notably, this work includes a lesion study that indicates the involvement of the basal ganglia area X in this form of learning.

My only suggestion for improvement of the manuscript is to include a more thorough discussion of how this work relates to citation #14, L. Veit's paper. The introduction and concluding sentence of the discussion claims novelty for the observation of songbirds volitionally modulating song content according to different contexts. My understanding is that this ability was already dramatically demonstrated in Veit et al. 2021. It would be great if the discussion and introduction could address the similarities and differences of the two studies more thoroughly. For example, the nature of the reward signal is distinct.

I find this paper a valuable contribution to the field and even though there is a lot of overlap with Veit 2021, the details of the two studies are different, particularly in the nature of the reward signal, and this new paper stands on its own as a significant novel contribution that provides further information about about the nature of real-time, reward-driven, context-dependent control of song syntax.

Reply: Thank you for the valuable comments. According to the suggestions, we have now included a paragraph in the Discussion section, describing about the relation to Veit's paper in detail. We believe our new Discussion highlights the significance of this study in addition to the contribution of Veit et al., in more concrete way. (lines: 267-286)

Reviewer #3 (Remarks to the Author):

In this paper, the authors present a set of results relating to the ability of adult Bengalese finches to modulate the sequential structure of their songs. They first describe an online algorithm that annotates syllables as they are produced by a singer in quasi real-time. They then use this annotation method to track the sequential structure of song as it's produced in an appetitive operant training procedure. Specifically, access to short videos of conspecifics is made contingent on the repetition of a target syllable a certain number of times. This training leads to an increase in the number of target syllable repetitions, and other operant manipulations (reward devaluation and extinction) modulate repetition rates in expected ways according to standard operant learning theory. When different reward contexts are cued visually, across either days or 1-hour blocks, repetition rates vary according to the cue (even over the short term in the absence of reward). The authors then report that lesions to the anterior forebrain pathway (Area X) reduce the learned syllable repetition rates and block subsequent context-dependent repetition changes. Finally, the authors identify target (non-repeat) syllable transitions and show that operant reinforcement can be used to modulate the frequency of these transitions as well.

Overall, there is a great deal to like in this paper. After reading it closely, I find myself generally sympathetic to the main conclusion that Bengalese finches can voluntarily modulate the sequential structure of their songs (at least to a limited degree). However, the paper is not well-written, the data are rather sloppily presented, and some of the conclusions made are not justified or are irrelevant. Thus, there are a number of serious shortcomings that preclude publication, at least in its present form. I detail the most significant of my concerns below:

Main effect of operant training; Figure 2:

- The various dependent measures plotted in panels c, d, e, g, h are not defined. One can intuit that the authors are trying to show changes in the number of repetitions in the operantly targeted syllable, but why is the same behavioral effect captured with five different measures that change (for no apparent reason) across the panels. This does not instill confidence in the stability of the results.*

Reply: We appreciate the reviewer for bringing this to our attention. In the initial manuscript, panels c and d illustrated the change of repetition number in within songs whereas panels e g h depicted the change of song number. The initial labeling of axes was indeed inconsistent and misleading; however, the change in song rate and frequency were actually identical. We have corrected the label and amended the presentation to be consistent (Fig. 2 and Supplementary Fig. 3).

- A paired -t-test is likely inappropriate for the data shown in Fig2C, as there is one bird that is clearly an outlier and differences may not be normally distributed. Similarly if we are supposed to compare the effect in 2c to the lack of an effect in 2d, then an LME model should be used.*

Reply: As the reviewer suggested we include statistical analysis based on linear mixed model based model where necessary. The results were now included in the revised main text (line: 119-126, 220-222, 250-252) and in Supplementary figure legend (Supplementary Fig.5)

- *Why are there data for 9 birds shown 2c and only 6 birds in 2e?*

Reply: We apologize for error. We have now included the data from all 9 birds (Fig. 2f).

- *This looks like classic operant shaping. They should make a note of this in the discussion.*

Reply: We have taken the feedback into account and clarified in our manuscript that our procedure correspond to shaping, as now outlined in (line: 135-136). We have also relocated the figure concerning “not too much” experiment in to Supplementary Figure. As mentioned latter, this was done because the “not too much” experiment were conducted on a specific population (n = 4) that extended the repetition of two syllable beyond the threshold following the reward conditioning (n = 9). We also correct our description that our initial statement “birds can maintain the syllable repetition within a specific range” was not appropriate when the results were interpret according to the shaping paradigm. We have moderated the statement to “birds could modulate the range of syllables repetition flexibly according to reward provided”, and moved to Supplementary Figure as the importance of this result lowered.

Relating to the main effects in figure 2: What exactly is the main hypothesis? Is it that any change in sequential structure that can be modified by operant learning is grounds for flexible control? Or are there only some forms of operant learning induced sequential changes that justify this claim. As it is the authors present several results showing various putative changes, some of which are stronger than others, and we are given no guide for how to interpret these in the context of the primary or secondary hypotheses, because no actual hypotheses are ever stated. The paper would benefit tremendously from a clear articulation of why particular manipulations and analyses are conducted.

Reply: We have included a sentence stating the hypothesis of our experiments as follows:

“We hypothesized that providing such social signals contingently upon the utterance of specific contents within the song would result in adaptive changes to the corresponding locus in their songs..” (line: 108-110)

The authors show no significant difference in the number of supra-threshold repetitions

($x+1 < R$) between “reward” and “not-too-much” reinforcement conditions (fig 2g, panel 3). This argues against the stated conclusion that birds could “... maintain them [syllable repetitions] within a specific range”, and instead supports the conclusion that the upper bound on the range is constrained by non-operant factors.

Reply: As the reviewer point out, we toned down the statement concerning the conclusion of this experiment. As a result, we moved this part of figure to Supplementary figure. The new conclusion are now below.

“These results demonstrate that birds could flexibly modulate the range of syllables repetition according to the reward provided. “ (line: 142-143)

It is difficult to interpret the results regarding “loci of modulation” without further experimentation. If the authors wish to claim that the locus depends on the “strategy” of each bird, then they should devise an empirical test (presumably involving selection of different targets within birds) to show that a proposed strategy is better than chance at explaining why a specific bird modulates the repetition at a specific locus. If a specific strategy cannot be identified, it is unclear what this section adds to the overall paper.

Reply: In this experiment, the specific loci of modulation is not the direct factor determining the reward outcome. Rewards were administered whenever the repetition at any locus surpassed the designated threshold (as depicted in original Method, line 589-591). We did not provide rewards based on the total number of repetition across all loci. As each bird was unaware of the rules governing the reward delivery, the locus modulated varied from bird to bird. We have included the more explanation about the hypothesis and the result of this section. (line: 156-159)

The overall statistical approach is weak:

I’ve pointed out a places where choice to use a paired t-test is not appropriate, but similar arguments might be applied to several other analyses, where authors treat select comparison as independent even when they involve the same subjects under different conditions. For example, in figure 6, panels f-g are not independent. Rahter than looking at two t-tests, an LME should be used to directly test the effect of the training. (Same for panels k-l). Likewise, when discussion the lesion effects, the author’s claims rest on within subjects comparisons across learning conditions (Fig 5d), but comparison to the control animals (shown only in supplementary Fig 4) is qualitative - i.e. there is an effect in one group but not the other. A LME model should be used to directly, and quantitatively compare the lesions and control birds.

Reply: As the reviewer suggested we include statistical analysis based on liner mixed model based model where necessary. The results were now included in the revised main

text (line: 119-126, 220-222, 250-252) and in Supplementary figure legend (Supplementary Fig.5)

The extent of the lesion in each animal is not quantified, but needs to be.

Reply: We included the extent of lesion in each animal. The results are now shown in the Methods (lines: 696-697)

Discussion:

The authors need to do more to differentiate their results from the existing literature which (as they correctly note) contains a large number of cases where aversive auditory feedback has been used to alter the singing behavior of different birds. Although the majority of these studies have focused on reward dependent modulation of spectral characteristics of song, it is not immediately clear that the kinds of effects observed here are necessarily distinct. One conservative interpretation is that the reported effects are just a versions of the reward mediated vocal learning that has already been widely reported.

Reply: We acknowledge this perspective. Many studies have uncovered learning-dependent or context-dependent modulation in the phonological aspect of syllables. We identified a previously overlooked phenomena, in addition to the results demonstrated in Veit et al., but never connected this finding to phonological modification. We have incorporated statements addressing this matter in the discussion g.

I would suggest dropping the “volitional” term from the title and the body of the paper. The strong assumption here is that an operantly trained behavior is equivalent to a voluntary one, but this is (and has been) a contentious point in the animal behavior literature for some time. Consider, for example, the case of addiction in which behaviors are acquired operantly, but which become involuntary. I am not sure that the present work supports any progress in this age old debate, and I don’t think that there is much to be gained by forcing this kind of wording on the reader.

Reply: Following a scrupulous reassessment, we have opted to follow the reviewer’s suggestion and adopt the more objective terminology, “goal-directed” instead of “volitional”.

As reviewer points out, discerning the voluntary aspect of behavior is challenging from mere observation, not only for animal even to human subjects as well. Nonetheless, our experiment was designed to fulfill the specific criteria required for the studying voluntary behaviors in clinical patients (Hopf HC et al., *Neurology*, 42:1918-1923, 1992; Cattaneo L et al., *Neurosci. & Behav. Rev.* 38:135-159, 2014). These criteria

have been employed previously in assessing the voluntary actions of songbirds in the studies conducted in Nieder's Lab (Brecht KF et al., *PLoS Biol.*, 17, e3000375, 2019; Brecht KF et al., *Cell Rep.*, 42, 112113, 2023). We think the specific example raised by the reviewer concerning an individual with addiction exhibiting involuntary behaviors, does not precisely align with the results of our experiment. This is because, even after the birds displayed learned behaviors, such as the elongation of repetition, they did not exhibit such tendencies on "No-reward" days when they were devoid of contingent reward delivery. Additionally, we demonstrated that diminishing the reward value led to a reduction in the repetition, further showing that the behavior occurred in a goal directed manner (Balleine BW, *Neuron* 2019). We have replaced the term "volitional" to "goal-directed" but have retained the term "volitional" in selected context where it is apt for discussion.

REVIEWER COMMENTS

Reviewer #1 (Remarks to the Author):

I would like to thank the authors for providing this revised manuscript. Most of my comments were properly addressed and I now consider this paper ready for acceptance.

Reviewer #3 (Remarks to the Author):

The authors have addressed most of my previous technical concerns regarding the methods and analyses.

The novelty of the current work is still not clear. Stating in the introduction that "it remains unclear whether birds can modulate the particular sequence of the songs..." is simply not true; They can. Based on prior published studies from a range of labs, we already know that songbirds can be operantly conditioned to modify their songs in many ways, including rapid context-dependent shifts in the sequential organization (Veit et al 2021). Likewise, we know that real-time live video feedback provides a powerful reinforcement signal during vocal learning (Carouso-Peck and Goldstein 2019). The present paper uses a less biologically relevant reinforcer (recorded videos) to shape slightly different kinds of sequential changes in Bengalese finch songs (syllable repetitions rather than transitions as in Veit et al). One could argue that there is some novelty in combining a positive reinforcer with the targeting of sequential structure, and I would agree with the statement (line 43, intro) that "whether [birds] can modulate the specific sequence in their songs to obtain social feedback is unknown". However, the current results are confirmatory, rather than revelatory, based on the published literature. The same is true for the devaluation and extinction components: the results are exactly as one would expect for any instrumentally conditioned appetitive behavior. Thus, claims in the abstract such as "we elucidate the flexibility of [sp.] songbirds exhibit in organizing and sequencing the order of syllables within their songs", and "Our analysis revealed that birds possess the capacity to modify the contents of their songs...", overstate the novelty of the current results, as do the various references to the goal-directed nature of the effects.

Similarly selective/loose framing of the current work in the context of the existing literature occurs throughout the abstract, introduction, and discussion. For example, the prior studies with white noise feedback include multiple species and more conditioned responses than just "refrain[ing] from uttering a particular sequence of syllables". In addition, the authors distort the prior work from Carouso-Peck, Goldstein (2019), to motivate the use of a "social" signal. What that study shows is that vocally contingent live video feedback is better than the same video stream when it's non-contingent, i.e. the content of the video (social or otherwise) was not the independent variable. It might be possible to frame a more specific (post-hoc) hypothesis regarding the content of the videos, but their limited nature in the present study is not very helpful. In the Carouso-Peck study, the real-time contingency of the feedback was shown to be very important for vocal learning. In the present study, the authors use a "movie with non-animate signals" as a control, then infer that "social reward" rather than "contingent visual signals" stimulate change in vocal structure. Given what we know from prior work, I suspect that they are correct, but I don't think this conclusion is supported by the study as there are many features in addition to the "social" nature that differ between the videos. (NB: social nature is never defined, but one infers that it is tied to the appearance of a conspecific). Ruling out the role of "contingent video" is fine as a baseline control, but it is not the same supporting the role of "social reward". Rigorous support for the claim as written requires further study examining multiple videos with various aspects, e.g. motion, species, sex and timing, tightly controlled.

So, behaviorally the only real novel, well-supported result is that syllable repetitions can be instrumentally modified. I don't know that this supports publication in nature communications. Beyond the behavior, the Area X lesions are interesting, but they are presented in a somewhat cursory way with only a main effect on the operant-induced changes described. Thus, we learn nothing about how the various components and aspects of this specific learning are mediated by/through the AFP, and so this does not take us very far. Again, this is not uninteresting from someone in the field, but I'm not sure it supports publication in nat comm.

New concerns:

Syllable annotation: no ground truth is established. Rather, Tweety Net is taken as the ground truth and SAIBS and human annotation are compared to that. They show that both TweetyNet and SAIBS differ from human annotation, but the direction of this difference is not clear. It is therefore misleading to describe the SAIBS decoder as "significantly more accurate" than manual annotation, as the claim is not supported. This does not necessarily pose a problem for the subsequent parts of the paper, as the parsing and labelling only needs to be consistent across songs. To the extent that it differs from ground truth (which is unknown), it does introduce noise into the subsequent training and online tracking. This might contribute to why the effects are tied to explicit syllables in each bird.

RESPONSE TO REVIEWERS' COMMENTS

Reviewer #1 (Remarks to the Author):

I would like to thank the authors for providing this revised manuscript. Most of my comments were properly addressed and I now consider this paper ready for acceptance.

Reply: We are deeply appreciative for the valuable comments. The comments provided by this reviewer have been instrumental in enhancing both the clarity and significance of our manuscript.

Reviewer #3 (Remarks to the Author):

The authors have addressed most of my previous technical concerns regarding the methods and analyses.

The novelty of the current work is still not clear. Stating in the introduction that “it remains unclear whether birds can modulate the particular sequence of the songs...” is simply not true; They can. Based on prior published studies from a range of labs, we already know that songbirds can be operantly conditioned to modify their songs in many ways, including rapid context-dependent shifts in the sequential organization (Veit et al 2021).

Reply: We value the perspective offered by the reviewer and are grateful for the opportunity to consider these insights. However, there are certain aspects of the feedback with which we respectfully disagree or find ourselves unable to concur. We would like to elaborate on these points to clarify our position and provide a more comprehensive understanding of our stance.

The reviewer’s assertion that “*it remains unclear whether birds can modulate the particular sequence of the songs...*” is not true” was challenged without offering prior evidence or references. It is important to clarify that our original statement in the introduction was specifically phrased as “it remains unclear whether bird can INTENTIONALLY modulate the particular sequence of their song...” (line 41–42). This statement references Brecht et al 2019 (ref 16), which demonstrated that crows can intentionally vocalize calls in response to experimentally provided cues. Their research establishes a tightly controlled experimental paradigm to illustrate that their behavior is volitional rather than affective. To date, no other experiment with other songbirds have been conducted to definitively discern whether songbirds vocalize in volitional or affective manner, as far as we know. In the study by Brecht et al., 2019, the focus was on crows, which typically produce short vocalization without a complex sequence. Their findings confirmed that crows can vocalize intentionally, but they did not address whether there is modulation in the sequence of syllables. Therefore, our statement, “whether bird can intentionally modulate the particular sequence of their song...” remains an open question prior to our current study.

The reviewer’s comment regarding the previous knowledge of song modification by operant conditioning experiment is inaccurate. Operant conditioning, a paradigm where specific behaviors lead to corresponding rewards or punishments, has traditionally been used in songbird research to assess visual or auditory recognition, often involving behaviors like key presses or movement within the cage. However, it’s important to distinguish between these traditional uses and the focus of our research, which assess the modification of vocalization of the subject. In this context, Tumer and Brainard in 2007 (ref 15) pioneered the approach that targeted the song itself as the operant behavior and using white noise as disruptive feedback (the phrase “disruptive” is used as in their paper). Their study observed changes in pitch—a phonological feature—rather than in the sequence of syllables. Subsequent studies from different laboratories (Andalman and Fee *PNAS* 2009; Canopoli et al., *J. Neurosci* 2014; Ali et al., *Neuron* 2013) have confirmed

these findings, neither of them observed significant change in sequence. Our approach contrasts with those works, as we do not use white noise feedback, and our primary focus is on syllable sequence rather than phonology. The subsequent research from the Brainard lab, specifically Warren 2012 (ref 13) and Veit 2021 (ref 14), marked the first observations of changes in the sequence of syllables in Bengalese finches. While we acknowledge these significant findings, our study extends this body of work by demonstrating changes in syllable sequence under a reinforcing conditions different from their work in variety of ways: positive feedback signals, non-disruptive feedback, and feedback mimicking natural communicative responses; and they occurred in volitionally rather than affectively. These differences and significance are discussed in detail in the discussion section. Considering the unique nature of our study and its deviation from traditional operant conditioning methods using white noise feedback, we felt a detailed comparison with these previous studies, except for those by Warren 2012 and Veit 2021, was not essential in the manuscript, especially at the aspect of our study highlighted by the reviewer regarding “refrain[ing] from uttering a particular sequence of syllables” (*line 102*), where the change in sequence is being discussed.

Likewise, we know that real-time live video feedback provides a powerful reinforcement signal during vocal learning (Carouso-Peck and Goldstein 2019). The present paper uses a less biologically relevant reinforcer (recorded videos) to shape slightly different kinds of sequential changes in Bengalese finch songs (syllable repetitions rather than transitions as in Veit et al). – omission – Again, this is not uninteresting fro someone in the filed, but I'm not sure it supports publication in nat comm.

Reply: The reviewer’s comment regarding our use of “*less biologically relevant reinforcer (recorded videos)*” “ in evoking sequential changes in Bengalese finch songs warrant clarification. The reviewer references Carouso-Peck and Goldstein 2019 (ref 23), which employed recorded video playback as contingent social feedback for song learning in juvenile zebra finches. This study revealed that such feedback stimulates song learning. It is crucial to note that the methodology in Carouso-Peck and Goldstein 2019 is similar to ours; they used recorded video playback (of a female bird), which was played contingently in response to the juvenile bird's song. Our study also used recorded video playback (of a conspecific bird), played contingently when the birds produced a song exceeding a pre-determined repetition threshold and including target syllable transitions (see Figs. 2–4, 6).

The reviewer appears to have misunderstood Carouso-Peck and Goldstein's methodology, mistakenly stating they used “*live video feedback*.” Like them, we employed recorded video playback, which is a standard practice in studies assessing the effect of “social signals” on behaviors like song learning and vocalization and others in various avian species (referencing studies from Keeling and Hurnik *Appl. Anim. Behav. Sci.* 1993; Adret et al, *J. Comp. Psychol.* 1997; Shimizu *Behav* 1998; Ikebuchi and Okanoya *Zool. Sci*, 1999; Mui et al *Proc R Soc B: Biol Sci* 2008; Deshpande et al., *Proc R Soc B: Biol Sci.* 2014; Guillette and Healy *Behav. Proc.* 2017; Ljubicic et al, *Curr. Opin. Behav. Sci.* 2016; James et al., *J. Exp. Bio.* 2019; Varkevisser et al., *Anim. Cong.* 2022). Actually, Carouso-Peck and Goldstein 2019 themselves concluded that “*crucial role of social feedback to immature vocalizations in the development of communication*”

(last sentence in their abstract), with their experiments which employed only video playbacks. Furthermore, our study extends the work of Carouso-Peck and Goldstein by demonstrating that song modulation is not observed with non-animate feedback (e.g., a circle picture), an aspect not explored in their research. This distinction highlights the importance of social content in feedback. Our findings support the conclusion that social feedback plays a crucial role in song learning and modulation, aligning with Carouso-Peck and Goldstein's conclusion about the importance of social feedback in vocal development.

The aim of our study is to demonstrate goal-directed song change in response to social feedback, not to assess the most effective content within such feedback. While further experiments to determine the most effective type of social feedback for song modulation would be valuable, it falls outside the scope of our current study. Our results do not “*distort*” but rather build upon the findings of Carouso-Peck and Goldstein 2019. In the revised manuscript, we now included a statement that similar contingent feedback of conspecific video have been utilized to stimulate song learning with citation of Carouso-Peck and Goldstein 2019 (*line 108–109*).

New concerns:

Syllable annotation: no ground truth is established. Rather, Tweety Net is taken as the ground truth and SAIBS and human annotation are compared to that. They show that both Tweety Net and SAIBS differ from human annotation, but the direction of this difference is not clear. It is therefore misleading to describe the SAIBS decoder as “significantly more accurate” than manual annotation, as the claim is not supported. This does not necessarily pose a problem for the subsequent parts of the paper, as the parsing and labelling only needs to be consistent across songs. To the extent that it differs from ground truth (which is unknown), it does introduce noise into the subsequent training an online tracking. This might contribute to why the effects are tied to explicit syllables in each bird.

Reply: It is important to note, contrary to the reviewer's understanding, that our comparison was not between human and SAIBS annotations. Our analysis involved comparing the accuracy of human and SAIBS annotations each against the TweetyNet annotations, which we treated as ground truth. This is already clearly written in our manuscript (*line 527–532, 538–543*). Specifically, we analyzed the congruence between TweetyNet and human annotations (72.27%) and between TweetyNet and SAIBS annotations (98.43%). The primary reason for the lower accuracy in human annotations is due to the inherent variability in visual inspection-based annotations. This variability is evidenced by the higher deviation observed in human annotations compared to those by SAIBS, and significant less coefficient variance (SAIBS 0.317%, manual 7.38%, *line 83–84*). The results clearly show that SAIBS is significant more accurate than manual annotation. Because the reviewer’s comment was based on false assumption that our comparison was between “human vs SAIBS” and “human vs TweetyNet”, we do not further modify our manuscript.

Regarding the impact of rare noise in annotations, the reviewer is correct in stating that these do not significantly affect the interpretation of our study's results. The high accuracy of the annotations overall (as indicated by the percentages, mean of 98.43%) ensures that rare mistakes do not substantially influence the outcomes. This is especially pertinent since the same syllable is consistently targeted in each bird throughout various experimental conditions tested, such as before and after reward conditioning, on Reward versus No-Reward days, and between Rule A versus Rule B conditions.

RESPONSE TO REVIEWERS' COMMENTS

Thank you very much for re-evaluating and providing on our manuscript (NCOMMS-23-21867B). We further addressed the comment of Reviewer #3 in more detailed way. Changes in the manuscript are highlighted with red color. Points of the revisions and our responses to the reviewer are detailed below:

Reviewer #3 (Remarks to the Author):

The first part of the reviewer's comment have been omitted because these have been already cleared in the last round of revision.

New concerns:

Syllable annotation: no ground truth is established. Rather, Tweety Net is taken as the ground truth and SAIBS and human annotation are compared to that. They show that both Tweety Net and SAIBS differ from human annotation, but the direction of this difference is not clear. It is therefore misleading to describe the SAIBS decoder as "significantly more accurate" than manual annotation, as the claim is not supported. This does not necessarily pose a problem for the subsequent parts of the paper, as the parsing and labelling only needs to be consistent across songs. To the extent that it differs from ground truth (which is unknown), it does introduce noise into the subsequent training an online tracking. This might contribute to why the effects are tied to explicit syllables in each bird.

Reply: Initially provided in our last round of revise (NCOMMS-23-21867B)

It is important to note, contrary to the reviewer's understanding, that our comparison was not between human and SAIBS annotations. Our analysis involved comparing the accuracy of human and SAIBS annotations each against the TweetyNet annotations, which we treated as ground truth. This is already clearly written in our manuscript (*line 527–532, 538–543*). Specifically, we analyzed the congruence between TweetyNet and human annotations (72.27%) and between TweetyNet and SAIBS annotations (98.43%). The primary reason for the lower accuracy in human annotations is due to the inherent variability in visual inspection-based annotations. This variability is evidenced by the higher deviation observed in human annotations compared to those by SAIBS, and significant less coefficient variance (SAIBS 0.317%, manual 7.38%, *line 83–84*). The results clearly show that SAIBS is significant more accurate than manual annotation. Because the reviewer's comment was based on false assumption that our comparison was between "human vs SAIBS" and "human vs TweetyNet", we do not further modify our manuscript.

Regarding the impact of rare noise in annotations, the reviewer is correct in stating that these do not significantly affect the interpretation of our study's results. The

high accuracy of the annotations overall (as indicated by the percentages, mean of 98.43%) ensures that rare mistakes do not substantially influence the outcomes. This is especially pertinent since the same syllable is consistently targeted in each bird throughout various experimental conditions tested, such as before and after reward conditioning, on Reward versus No-Reward days, and between Rule A versus Rule B conditions.

Reply: Added in response to the editor's instruction

The reviewer's concern regarding the appropriateness of TweetyNet as the "ground truth" for syllable annotation merits consideration. However, we believe this discussion might not be particularly productive because of following reasons. The true "ground truth" of bird song annotation is, in fact, inaccessible to human observer through any objective methods. This is because communicative signals have a subjective nature, and how these signals are recognized can only truly judged by the birds that use them. Thus, the standard objective way of annotating birdsong relies on the visual inspection of a syllable's spectrogram, but this do not necessarily tell the true annotation. This discrepancy arises because the bird's auditory recognition does not necessarily follow an even distribution of spectral power across frequency spectrum, a premise upon which typical spectrogram based syllable labelling is often based. Consequently, what the songbird researchers strive to do is to precisely match the spectral feature of the sound to specific syllable identities and label them as accurately and consistently as possible. Therefore, a direct comparison between manual labeling and SAIBS labeling is technically meaningless, as no method can definitively identify the "ground truth" of bird song annotation. For this reason, we refrain from discussing the accuracy score or match rate of human versus SAIBS comparison. For quantitative and statistically compare the two methods, we hypothetically set the correct answer to be the one created using the third party created TweetyNet. We do not consider TweetyNet the "ground truth" of birdsong annotation; instead, we use it as a reference for measuring the accuracy and consistency of syllable annotation by human and SAIBS. Since it's publication, TweetyNet is considered as one of the most accurate syllable annotator for birdsong and used for evaluating song annotation of other methods in various studies (Cohen et al., *Nature* 2020; Steinfath et al., *eLife*, 2021; Provost et al., *PlosONE* 2022; Yang et al., *Ornithology* 2023; Martin *Ecological Informatics* 2022; Koparkar et al., *bioRxiv* 2023). Although the correctness of its annotation is well appreciated, since it is an offline decoder, this program can not be applied in the online rapid decoding of ongoing stream of audio signals that is required in this study. Thus we created SAIBS, and evaluated the accuracy of it comparing with TweetyNet. Contrary to the reviewer's understanding as revealed by the statement of "*They show that both Tweety Net and SAIBS differ from human annotation*", our comparison was not conducted between human annotation versus SAIBS annotations. Our analysis involved comparing the accuracy of human and SAIBS annotations each against the TweetyNet annotations. Since TweetyNet requires annotated template for learning, we used the annotation created automatically by SAIBS. We consider this "TweetyNet trained with SAIBS created annotation" as the reference for comparison, and compared the accuracy of

“TweetyNet trained with SAIBS created annotation” versus manual annotation (Supplementary Fig. 1a) and “TweetyNet trained with SAIBS created annotation” versus SAIBS online decoder (Supplementary Fig 1b). The sequence identity rate for each researcher was compared to TweetyNet. The result from four researcher were $72.27 \pm 5.33\%$. Then we performed the same process with three independently trained SAIBS and got the score of $95.28 \pm 0.30\%$. This was statistically significant with $P = 4.84 \times 10^{-3}$ by Welch’s t-test (line 78). The primary reason for the lower accuracy in human annotations is due to the inherent variability in visual inspection-based annotations. This variability is evidenced by the higher deviation observed in human annotations compared to those by SAIBS, and significant less coefficient variance (SAIBS 0.317%, manual 7.38%, line 83–84). The results clearly show that SAIBS is significant more accurate than manual annotation and can be considered to be equivalent to TweetyNet as an online decoder. In the initial review, we have added description to explain the rational for such comparison in the methods section. (line: 536-539).

Regarding the impact of rare noise in annotations, the reviewer is correct in stating that these do not significantly affect the interpretation of our study's results. This is because the core issue in the operant learning paradigm requires consistently providing the reinforcer across the behavioral epoch (Skinner 1958). This was accomplished by the high accuracy of the annotations overall (as indicated by the percentages, which is 98.43%, line: 73) against TweetyNet, and high consistency across the redention (98.26%, line: 81), ensuring that rare mistakes do not substantially influence the outcomes. Moreover, the consistent provision of the reinforce signal is especially warranted since the same syllable, the repetitive syllable in the most part, is consistently targeted in each bird throughout various experimental conditions tested, such as before and after reward conditioning, on Reward versus No-Reward days, and between Rule A versus Rule B conditions. In addition, the targeted syllable was different for each bird because each birds use different pattern of syllables (line: 599-601), but we observed consistent result across the subjects (Figs 2c-f, Figs 4 b-h). The “noise” mentioned by the reviewer corresponds to annotation error that results in either a false negative – not providing the reinforcer even thought the song met the criteria, or a false positive – providing the reinforcer even thought the song did not meet the criteria. A false positive is highly unlikely because it is virtually impossible for an unrelated sequence of syllable to meat the criteria, such as repetition over a certain threshold, by chance. A false negative may occur if SAIBS mistakenly annotates the last few syllable in a repetitive syllable as a different syllable, which may be calculated to occur in 1.57% of song bout for a single error and 0.024% for a double syllable annotation error. Even though such error happen in a consistent manner for every song during reinforcement, this noise does not result in changing the song, because the reinforcer was not provided in such cases. In this study, we observed the opposite, clear increase in the repetition, which is unlikely occur by annotation error. Thus, it can be concluded that these rare errors do not significantly affect the interpretation of our results. This reviewer’s speculation “*This might contribute to why the effects are tied to explicit syllables in each bird.*” requires clarification. Since our target of conditioning is on a single syllable for each bird, expect for the experiments shown in Figure 6 where a

two- pattern of syllable sequence was targeted for each bird, and the target syllable differed to each bird (*line: 599-601*), we do not show any data that suggest that changes in song occurred depending on the syllable target we selected. The target was determined with clear criteria, the repetitive syllable or branching sequence having similar similarity score in word2vec analysis as already described in the methods (*line: 599-601, 678-680*). We further demonstrated that although the same syllable was targeted for conditioning training, no change in song was observed in response to non-social feedback (Fig. 2e), and a consistent decrease in repetition was noted on excess reward days (Fig. 4b) and in no-reward conditions (Fig. 4d). This challenges the speculation that the change in sequence were due to annotation errors from the “ground truth” annotation which is inaccessible for human observer.

In the revised manuscript, we have added a statement to explain why we compare the accuracy of annotation by human and SAIBS against TweetyNet. This includes references to published studies that evaluated the accuracy of TweetyNet. Additionally, we have detailed the values resulting from this comparison, previously described in the main text, now also in the method section (*line: 550-557*).